# VISUAL PERCEPTION IN TEXT STRINGS

## ABSTRACT

Understanding visual semantics embedded in consecutive characters is a crucial capability for both large language models (LLMs) and multi-modal large language models (MLLMs). This type of artifact possesses the unique characteristic that identical information can be readily formulated in both texts and images, making them a significant proxy for analyzing modern LLMs' and MLLMs' capabilities in modality-agnostic vision understanding. In this work, we select ASCII art as a representative artifact, where the lines and brightness used to depict each concept are rendered by characters, and we frame the problem as an ASCII art recognition task. We benchmark model performance on this task by constructing an evaluation dataset with an elaborate categorization tree and also collect a training set to elicit the models' visual perception ability. Through a comprehensive analysis of dozens of models, results reveal that although humans can achieve nearly 100% accuracy, the state-of-the-art LLMs and MLLMs lag far behind. Models are capable of recognizing concepts depicted in the ASCII arts given only text inputs indicated by over 60% accuracy for some concepts, but most of them achieves merely around 30% accuracy when averaged across all categories. When provided with images as inputs, GPT-4o gets 82.68%, outperforming the strongest open-source MLLM by 21.95%. Although models favor different kinds of ASCII art depending on the modality provided, none of the MLLMs successfully benefit when both modalities are supplied simultaneously. Moreover, supervised fine-tuning helps improve models' accuracy especially when provided with the image modality, but also highlights the need for better training techniques to enhance the information fusion among modalities. All resources are available at `https://anonymous.4open.science/r/VisionInText-08D3`.

## 1 INTRODUCTION

While conventional wisdom suggests that texts primarily function as carriers of linguistic information and images as conveyors of visual information, real-world scenarios often involve the integration of multiple information formats. For example, images may carry textual information, thus Optical Character Recognition (OCR) (Mori et al., 1992) has been extensively studied. It focuses on capturing and understanding linguistic information embedded in images through visual processors, which is a crucial ability required in modern models for visual reasoning tasks (Yu et al., 2023).

In contrast, the comprehension of visual information embedded within text strings has not received commensurate attention. One representative example that reflects visual semantics by a sequence of characters is ASCII art (Xu et al., 2016) as shown in Fig. 1. Visual information in these artifacts is situated in the middle of text strings and images, and can be readily expressed in both formats containing identical content. In other words, it is modality-agnostic.

Understanding how well models can capture visual semantics in text modality is significant for developing large language models (LLMs) (Dubey et al., 2024; Bai et al., 2023a). Upon pre-training on a vast amount of text corpus, language models are capable of capturing visual information through escape characters, such as "\n" and "\t", which encodes 2D structures in human writings. However, they were predominately assessed via textual-semantic-based evaluation benchmarks, without detailed analysis on its visual perception ability. ASCII art, where information can be fully represented in text strings, serves as an ideal tool for benchmarking LLMs' visual perception ability.

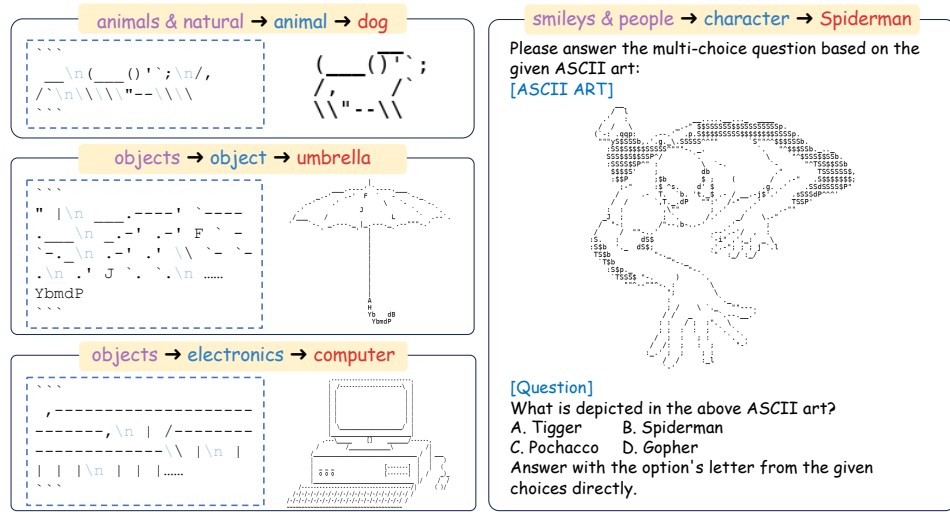

Figure 1: Examples of ASCII art. The left side contains text and image modalities of ASCII art pieces under different categories, where the texts are reformatted and truncated due to space limitation. The right side presents a multiple choice question in ASCIIEVAL.

Besides, with the advent of multi-modal large language models (MLLMs) (Achiam et al., 2023; Reid et al., 2024; Anthropic, 2024) that arm LLMs with visual processors, the aforementioned modality-agnostic characteristic also naturally leads to a new perspective of understanding MLLMs. The modality-agnostic feature of ASCII art ensures that both vision and text modalities have the identical semantics, which encounters the strict requirements for evaluating cross-modality alignment. In other words, we expect that MLLMs can not only perform robustly among different modalities, but also take the best of both worlds when two modalities are presented simultaneously.

Moreover, this research can also benefit a wide range of applications and have significant safety implication for LLMs and MLLMs. Such visual information is ubiquitous in a wide range of practical scenarios, such as processing tabular data (Deng et al., 2024) and playing board games (Topsakal & Harper, 2024). In addition, using visual information reflected in characters to break through the defense line is becoming a threat to LLM safety issues (Jiang et al., 2024b). For example, the attacker may use the ASCII art of a "bomb" instead of the word. A thorough analysis for understanding models' visual perception ability to make proactive defense is in urgent need.

In this work, we define ASCII art recognition as an ideal proxy to investigate models' visual perception ability in text strings through comprehensive evaluation and fine-tuning. Different from previous work that has focused on box diagrams (Hayatpur et al., 2024; Bayani, 2023), rich-formatting texts (Jiang et al., 2024b), or tone-based ASCII art (Wang et al., 2023a) that can be easily generated by rules or converted from images, we focus on ASCII art drawn by human artists, which is notably more abstract, replete with visual information, and more popular among people. We formulate the task as a multiple-choice question-answering problem, where the answers are objective for straightforward verification, to achieve fairer comparisons. Then, we task models to recognize the concept depicted in the ASCII art. Due to the lack of a dataset covering diverse categories thoroughly benchmarking the ability of existing models, we crawled data from online websites and cleaned manually under an elaborate categorization tree. In this way, we construct a test set dubbed ASCIIEVAL covering 359 concepts. To further elicit the models' visual perception ability, a training set was collected with approximately 10k data points.

We convert each ASCII art into a text string, an image, or both modalities at the same time as inputs, evaluated dozens of existing LLMs and MLLMs, and fine-tuned representative open-source models. Our major findings are summarized as follows:

   ◦ Models can truly recognize visual semantics through text inputs, indicated by the over 60% accuracy of GPT-4o in certain concept categories. However, existing LLMs performs poorly on ASCIIEVAL, where most of them achieve merely around 30% accuracy (Sec. 5.1).

○ There is an oversight in modality alignment that hinders MLLMs from answering questions flexibly among modality-agnostic visual signals. We observed that well-known MLLMs show a strong bias towards image modality, the expected synergistic effects do not emerge, and their training techniques fail to facilitate the backbone LLMs' visual understanding ability (Sec. 5.2 & 5.3.1).

○ LLMs and MLLMs show different trends in model performance when provided with different input modalities and excel at different ASCII art categories. Specifically, they perform relatively better on ASCII art containing fewer characters when given text inputs, whereas performing better on those with more characters given image inputs (Sec. 5.3.2 & 5.3.3).

○ Better training strategies or model architectures are required for optimizing modality-agnostic visual perception in text strings. Supervised fine-tuning using task-specific training data helps MLLMs leverage representations from different modalities slightly better, but shows little improvement given only text inputs (Sec. 5.4).

## 2 Backgrounds & Related Work

### 2.1 LLM & MLLM Benchmarks

Mainstream evaluations for LLMs focus on abilities in world knowledge, common sense reasoning, instruction following, long context modeling, and mathematical reasoning. Representative benchmarks include MMLU (Hendrycks et al., 2020), C-Eval (Huang et al., 2024), GSM8K (Cobbe et al., 2021), and StrategyQA (Geva et al., 2021). Except for recent work from Qiu et al. (2024) benchmarking LLMs on answering questions related to the graphics content by generating programs, none of them consider the visual perception ability of LLMs as a distinct research problem.

Benchmarks for MLLMs focus on similar abilities when given a mix of text and images, such as MMMU (Yue et al., 2024), MMBench (Liu et al., 2023b), and MME (Yin et al., 2023). Most images considered in these benchmarks are photographs, paintings, or comics, rather than visual information reflected in text characters. Additionally, the information between interleaved images and texts is not guaranteed to be equivalent or complementary, whereas information in ASCII art can be semantic-equivalent among different modalities.

Current benchmarks contain some ASCII art-related tasks. For example, Gu et al. (2024) introduces a fine-grained and diverse instruction-following evaluation dataset, in which ASCII art generation is a single case with approximately 40 samples of varied user requests. BigBench (Ghazal et al., 2013) contains ASCII MNIST digit recognition, ASCII word recognition, and ASCII kanji recognition. All of these tasks challenge the LLMs in recognizing different characters within the ASCII art. Test cases can be easily collected by using automatic conversion tools like Figlet [1], where models may learn conversion rules instead of truly understanding visual semantics.

In contrast, our work focuses on ASCII art depicting real-world profiles, containing more abstract visual features. We consider ASCII art recognition to be a preliminary ability for ASCII art generation, and propose ASCIIEVAL based on this task, which can serve as both an LLM benchmark and an MLLM benchmark, bringing unique characteristics compared to existing benchmarks.

### 2.2 Research on ASCII Arts

The history of ASCII art can be traced back to the 1860s. Due to the limitations of early computers, text characters were widely used to simulate graphs, gradually becaming an important graphic design technique. ASCII art broadly includes diverse types and styles (Carlsson & Miller, 2012; Carlsson, 2017), such as line art, emoticons, colored ASCII art, and animated ASCII art. Strictly speaking, it refers to art made up of 95 printable fixed-width ASCII characters (Xu et al., 2016), which are easy to copy from one file to another and display consistently across different computers.

Early studies focused on extracting ASCII art from general texts (Hiroki & Minoru, 2005; Hayashi & Suzuki, 2009; Suzuki, 2011) by exploring byte patterns, morphological features, compression ratios, etc. Subsequently, ASCII art gained more attention in the area of computer vision. Researchers generally categorize ASCII art into tone-based and structure-based types and have developed algorithms to synthesize ASCII art from images (Xu et al., 2010; Takeuchi et al., 2013; Xu et al., 2016;

---

[1] https://en.wikipedia.org/wiki/FIGlet

Chung & Kwon, 2022). Tone-based ASCII art emphasizes the intensity distribution of the reference image, while structure-based ASCII art focuses on the major structure of the content. The latter is mostly curated by human artists and is more challenging to synthesize automatically.

Works on ASCII art classification typically convert such text graphics into images as a default setting and exploit different image features to improve the classification accuracy of deep neural networks (Fujisawa et al., 2020a; Matsumoto et al., 2018; Fujisawa et al., 2018). Fujisawa et al. (2020b) constructs ASCII art data automatically to enhance the models' image classification ability. Most of the aforementioned works are tested using an ASCII art classification dataset containing only five categories, which is inadequate for comprehensively analyzing how well the LLMs and MLLMs can grasp the visual representation of ASCII art.

There are also works that take advantage of ASCII art to achieve specific goals. Jiang et al. (2024b) represent rich-formatting texts as ASCII art and find that it results in highly effective jailbreak attacks that bypass state-of-the-art defense techniques. In contrast, Wang et al. (2023a) find that tone-based ASCII art with rich visual information cannot be understood by current LLMs, which can be used as an effective tool to detect whether the participant is a bot or a human. ASCII art is also utilized to enhance LLMs' spatial reasoning ability in Wu et al. (2024)'s work. Box diagrams, as a special kind of ASCII art, are widely used in the development lifecycle (Hayatpur et al., 2024) and have been benchmarked by Bayani (2023) with recognition and generation tasks.

In this work, we regard ASCII art as an ideal information carrier that bridges the gap between the text and image modalities, to facilitate the understanding of modality-agnostic visual perception ability for both LLMs and MLLMs.

## 3 ASCII Art Recognition

We first define the ASCII art recognition task formally. Then, we introduced how we constructed the test and training data, dubbed ASCIIEVAL and ASCIITUNE, followed by statistical analysis.

### 3.1 Problem Formulation

We formulate ASCII art recognition as a multiple-choice question-answering problem. Let $T$ represent the text string of an ASCII art and $I$ refer to the corresponding image modality. The model is asked to predict the correct choice containing the concept depicted by $T$ or $I$ among candidates $C$.

For LLMs that only accept text input, the prediction $\hat{y}$ is generated as follows:

$$\hat{y}_T = \text{LLM}(T, C) \tag{1}$$

For MLLMs, $\hat{y}$ can be inferred under two additional conditions:

$$\hat{y}_I = \text{MLLM}(I, C)$$
$$\hat{y}_{IT} = \text{MLLM}(I, T, C) \tag{2}$$

We denote the above three input conditions as Text-only, Image-only, and Text-Image, respectively. The final prompt is structured by corresponding string templates given the inputs. See Appendix C.

### 3.2 Data Collection for ASCIIEVAL

We carried out the data construction process in four stages to collect a high-quality test dataset.

**Data Preparation** We first crawled ASCII art created by human artists from two online galleries [2].

**Classification Criteria Unification** Next, we manually designed a *3-layer classification tree* after unifying the categories based on the categorical information from the original websites and removing potentially harmful categories. The most fine-grained category is named the **concept**, representing the semantic meaning reflected in the art. Similar concepts are merged into second-layer **groups**. Finally, they are grouped into seven major **classes** inspired by the iOS emoji categories. Each concept can be depicted in various ways by ASCII artists.

---

[2]https://asciiart.website/, https://www.asciiart.eu/

**Data Filtering** Subsequently, we conducted additional filtering operations using a combination of rules and human annotations as follows:

○ Each ASCII art string was normalized by removing redundant empty spaces at the beginning of each line and at the end of the string, without compromising its visual semantics.

○ ASCII art consisting of more than 100 lines, not belonging to reserved categories, and repetitive to other ASCII arts under the same concept were discarded. Repetition was identified by calculating the edit distance between two ASCII strings. If the distance divided by the length of the existing string was smaller than 0.3, the new ASCII art will be considered redundant.

○ Human annotators were tasked to filter out unrecognizable or ambiguous art, remove words in ASCII art to focus the dataset on visual perception and avoid information leakage through words, and adjust the category according to the 3-layer category tree (See more analysis in Appendix D).

**Multiple-Choice Data Construction** Finally, we collected negative choices for each ASCII art by randomly sampling from other concepts within the same group. It should be noted that the ground truth labels were initially collected from the websites and subsequently verified by human annotators during the data filtering process. Each ASCII art string was then converted into an image.

## 3.3 Data Collection for ASCIITune

To further elicit models' visual perception ability through supervised fine-tuning on the ASCII art recognition task, the creation of a training set is essential. An intuitive solution is to leverage previous works on ASCII art synthesis (Xu et al., 2016; 2010) by converting existing image datasets, such as ImageNet (Deng et al., 2009), into the required format. A public dataset [3] indicates that after automatic tone-based synthesis, approximately 85% data samples are filtered out due to poor quality. Furthermore, existing data conversion tools are inadequate for structure-based ASCII art, which accounts for 94% of the data in ASCIIEVAL according to annotators' labels. Additionally, artists often combine both tone-based and structure-based features in a single artifact.

Therefore, we chose to collect the training set in a manner similar to ASCIIEVAL instead of relying on automatic conversion. Data sources include ASCII arts from another less well-organized website [4], and the crawled content was extracted into individual ASCII art pieces based on specific rules derived from observations. We also included the unrecognized ASCII art that was previously withdrawn during the construction of ASCIIEVAL. The normalized ASCII art is discarded if recognized as repetitive with samples in ASCIIEVAL or among each other.

Due to the large amount of data with diverse concepts, carefully categorizing data for high-quality distractors is unfeasible. Instead, we prompted Llama-3-70B-Instruct to generate negative choices given the ground truth concept and utilized the Perspective API to filter out unsafe samples based on the concatenation of candidate choices. Samples with scores less than 0.2 across all six dimensions, i.e., toxicity, severe toxicity, identity attack, insult, profanity and threat, are retained.

## 3.4 Data Analysis

As shown in Table 1, ASCIIEVAL comprises 3,526 samples distributed across 359 concepts, 23 groups, and 7 classes. The data distribution is illustrated in Fig. 2 (More in Appendix D). Each concept is represented by 9.82 ASCII art pieces on average, with a maximum of 170 and a minimum of 1, indicating an imbalance. ASCIITUNE consists of 11,836 samples with 2,307 concepts, which is more diverse but of lower quality. The number of characters and lines in ASCIIEVAL range from 4 and 1 to 15,282 and 100, respectively, reflecting its diversity and complexity. ASCIITUNE holds similar statistics.

**Human Upper Bound** We randomly extracted 100 samples from ASCIIEVAL three times and asked three different annotators to perform the multiple-choice task. They achieved 100%, 98% and 97% accuracy, respectively, demonstrating that this task is simple for humans.

---

[3] https://huggingface.co/datasets/mrzjy/ascii_art_generation_140k
[4] https://ascii.co.uk/art

Table 1: Statistics of ASCIIEVAL and ASCIITUNE.

| Dataset | | ASCIIEVAL | ASCIITUNE |
|---|---|---|---|
| #Samples | | 3,526 | 11,836 |
| #Concepts | | 359 | 2,307 |
| #Characters | Min | 4 | 1 |
| | Max | 15,282 | 13,569 |
| | Avg | 635.53 | 622.38 |
| #Lines | Min | 1 | 1 |
| | Max | 100 | 97 |
| | Avg | 16.97 | 15.22 |

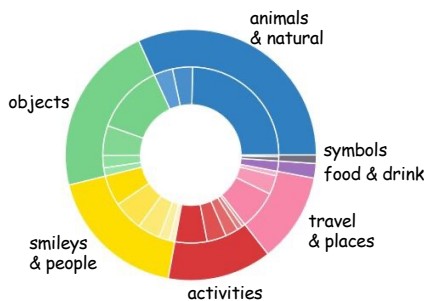

Figure 2: Data distribution of ASCI-IEVAL. The outer and the inner circle represent different classes and groups.

## 4 EXPERIMENT SETUP

### 4.1 EVALUATED MODELS

For open-source instructed models, we experiment with LLMs from different model families, including **Llama** (Touvron et al., 2023), **Qwen** (Bai et al., 2023a), **Mistral** (Jiang et al., 2024a) and **Gemma** (Team, 2024b), and with MLLMs from **Llava** (Liu et al., 2023a), **CogVLM** (Wang et al., 2023b), **Qwen-VL** (Bai et al., 2023b) and **Chameleon** (Team, 2024a). Besides, **GPT-4o** (OpenAI, 2023) and **Gemini** (Reid et al., 2024) are selected as two leading proprietary models. Both of them are multi-modal models capable of accepting text or image inputs and outputting text. The specific versions we used are gpt-4o-2024-05-13 and Gemini-1.5-pro. More in Appendix E.

All of the models are decoded using greedy search or by setting the temperature to 0 for fair comparisons and easier reproduction. The maximum number of output tokens equals 32 for open-source models and 128 for proprietary ones.

### 4.2 EVALUATION METRICS

We perform an exact match between the correct option and a model's output to calculate *accuracy* on ASCIIEVAL. As analyzed in Sec 3.4, the test data is unbalanced with varying art counts under each concept. Therefore, we adopt *micro-accuracy* over each sample for analyzing specific ASCII art characteristics, and *macro-average* over each concept for quantifying model performance. We also define *pass rate* to measure a model's ability to successfully follow the instruction by providing an effective answer. Proprietary models may fail due to their safety policy.

## 5 RESULTS AND ANALYSIS

In this section, we first benchmark the performance of LLMs and MLLMs on ASCIIEVAL. Next, we investigate whether the existing MLLM training approaches enhance the vision understanding abilities of LLMs and delve deeper into understanding which types of ASCII arts are more challenging. Finally, we examine whether supervised fine-tuning can better align models for this task.

### 5.1 PERFORMANCES OF LLMS

The performance of LLMs with only text inputs is shown in Fig. 3. Most of these models exhibit strong instruction-following abilities, achieving a pass rate equaling 100%. Therefore, this metric is not shown in the figure and we primarily focus on macro-accuracy comparisons.

**Proprietary Models vs. Open-source Models** GPT-4o performs best among all the models. It outperforms the best open-source model, Gemma-2-27B-it, by 32.51%, indicating a conspicuous gap between the leading proprietary models and open-source ones. Gemini ranks second, outperforming the open-source models, but still significantly lags behind GPT-4o.

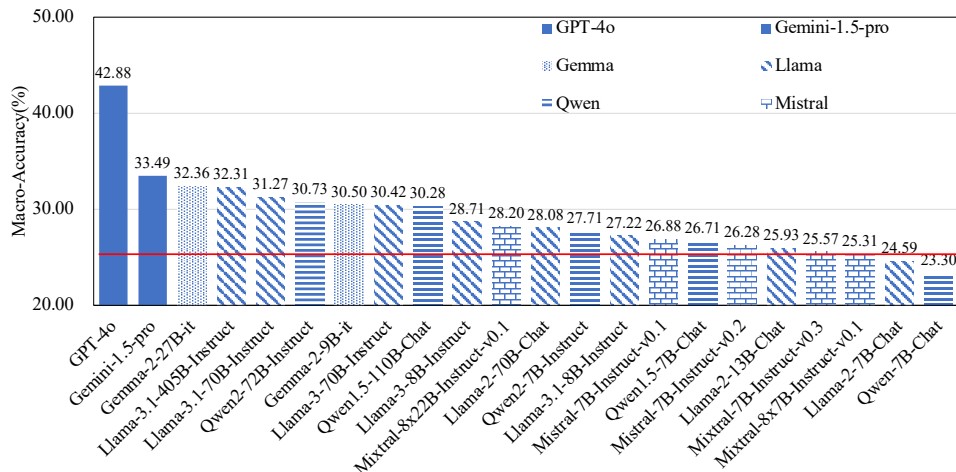

Figure 3: Macro-accuracy of LLMs on ASCIIEVAL. The red line is the random baseline (25%).

**Comparisons among Model Families** Models within the same series generally exhibit performance proportional to their sizes. However, this trend does not hold true across different model series and families. For example, Qwen2-72B-Instruct outperforms Qwen1.5-110B-Chat. Additionally, Gemma, with only 27B parameters outperforms other competitors with more than 70B and even hundreds of billions of parameters. This underscores the potential of developing lightweight models with strong visual perception abilities in text strings.

**Overall Performances of LLMs** Most models with fewer than 10B parameters, including the MoE model Mistral-8x7B-Instruct-v0.1, perform similarly to a random baseline. None of these models achieve an accuracy higher than 50%, with GPT-4o ranking first at only 42.77%. Although it's hard to guarantee that the ASCII art in ASCIIEVAL was never used during pre-training, the poor accuracy reflects that ASCIIEVAL stands as a challenging benchmark for LLMs, underscoring the oversight of visual perception ability in current LLMs.

## 5.2 PERFORMANCES OF MLLMs

We evaluate MLLMs using different input modes as introduced in Sec. 3.1.

Table 2: Performance of MLLMs with different input modalities. Accuracy (%) refers to macro accuracy. Pass rate (%) is listed to show the instruction-following ability of MLLMs. The highest accuracy is in bold and the second highest are underlined. Models are ranked by Avg, defined as the mean of the accuracy under different modes horizontally.

| Models | Avg | Text-only | | Image-only | | Text-Image | |
| --- | --- | --- | --- | --- | --- | --- | --- |
| | | Accuracy | Pass | Accuracy | Pass | Accuracy | Pass |
| GPT-4o | **67.36** | **42.88** | 99.97 | **82.68** | 98.75 | **76.52** | 99.83 |
| CogVLM2-Llama3-chat-19B | 53.07 | 24.73 | 99.32 | 67.80 | 100 | 66.68 | 100 |
| Llava-v1.6-34B | 51.87 | 28.62 | 100 | 65.66 | 100 | 61.33 | 100 |
| Gemini-1.5-pro | 50.84 | 33.49 | 97.36 | 60.69 | 99.46 | 58.33 | 98.78 |
| Llava-v1.5-13B | 49.52 | 26.00 | 100 | 61.87 | 100 | 60.70 | 100 |
| Llava-v1.5-7B | 49.45 | 24.66 | 100 | 62.18 | 100 | 61.52 | 100 |
| Llava-v1.6-mistral-7B | 48.54 | 25.89 | 100 | 60.72 | 100 | 59.02 | 100 |
| Llava-v1.6-vicuna-13B | 47.43 | 26.03 | 100 | 59.70 | 100 | 56.55 | 99.52 |
| CogVLM-Chat-hf | 46.61 | 21.25 | 86.07 | 61.00 | 100 | 57.58 | 99.97 |
| Qwen-VL-Chat | 39.10 | 24.79 | 90.70 | 52.32 | 96.68 | 40.09 | 77.94 |
| Chameleon-30B | 21.08 | 0.01 | 3.29 | 34.54 | 99.97 | 28.70 | 100 |
| Chameleon-7B | 18.13 | 0.00 | 0.00 | 26.46 | 96.17 | 27.93 | 99.40 |

**Proprietary Models vs. Open-source Models** Results in Table 2 indicate the gap between GPT-4o and other models for MLLMs. It achieves 43.88%, 82.68%, and 76.54% on the three modes with nearly 100% pass rate, while the second-best results lag behind by 14.26%, 14.86%, and 9.84%

in accuracy respectively. GPT-4o not only handles character strings better but also understands the ASCII art images well regardless of the style and abstractiveness differences compared to other image datasets, such as ImageNet (Deng et al., 2009) and MS-COCO (Chen et al., 2015). Nevertheless, GPT-4o, with the most competitive setting, still underperforms the human upper bound (98.33%).

**Comparisons among Input Settings**  Another observation is that the performance follows the trend Image-only > Text-Image > Text-only. Image encoders in MLLMs capture the visual information in text strings more effectively, leading to superior performance over the Text-only mode. Generally, we expect that multi-modal models can provide a more holistic understanding of the data. However, when incorporating text modality with image, the performance of all models except Chameleon-7B drops with a maximum decrease of 12.32% compared to the Image-only setting. This reveals that existing MLLMs are unable to understand the complementarity and consistency of different modalities, resulting in an inability to make correct predictions.

**Degradation of Instruction-following Ability**  We also observe that the instruction-following ability of some MLLMs, as indicated by the pass rate, drops significantly when ASCII art is provided in text strings. Among open-source late-fusion MLLMs, models from the Llava family show little influence on the backbone LLMs' ability, whereas others experience considerable degradation. The only early-fusion MLLMs, Chameleon, falls to approximately 0% accuracy and pass rate under the Text-only setting. Ideally, early-fusion strategies should better integrate the representations and interactions among modalities, leading to a more cohesive and accurate understanding of data. However, their poor performance and notable decline indicate significant room for improvement.

## 5.3 ANALYSIS OF THE RESULTS

As most open-source MMLMs are trained on a pre-trained LLM using late fusion strategies, we first investigate whether the LLM's visual perception ability improves after MLLM training. Next, we analyze the trends in model performance under different ASCII art sizes and categories. The top 5 LLMs under the Text-only setting and the top 5 MLLMs under the Image-only setting, which are generally their default input modalities, are primarily considered.

### 5.3.1 DO LLM'S VISUAL PERCEPTION ABILITY EVOLVE AFTER MLLM TRAINING?

Previous work on multi-modal models usually focuses on MLLMs' visual understanding ability over LLMs. A natural question arises: *Can an MLLM's training under the late fusion strategy enhance its backbone LLM's visual perception ability?* Given the semantic-equivalence feature of ASCII art, this question potentially explores how well the representations among different modalities are fused.

Table 3: Comparisons of MLLMs and their backbone LLMs measured by macro-accuracy (%).

| MLLM | LLM (backbone) | MLLM Acc | LLM Acc | $\Delta$ |
|------|----------------|----------|---------|----------|
| Llava-v1.5-7B | Vicuna-v1.5-7B | 24.66 | 26.05 | -1.39 |
| Llava-v1.5-13B | Vicuna-v1.5-13B | 26.00 | 25.47 | 0.53 |
| Llava-v1.6-mistral-7B | Mistral-7B-Instruct-v0.2 | 25.89 | 26.28 | -0.39 |
| Llava-v1.6-vicuna-13B | Vicuna-v1.5-13B | 26.03 | 25.47 | 0.56 |
| Llava-v1.6-34B | Nous-Hermes-2-Yi-34B | 28.62 | 27.88 | 0.74 |
| CogVLM-Chat-hf | Vicuna-v1.5-7B | 21.25 | 26.05 | -4.80 |
| CogVLM2-Llama3-chat-19B | Llama-3-8B-Instruct | 24.73 | 28.71 | -3.98 |
| Qwen-VL-Chat | Qwen-7B-Chat | 24.79 | 23.30 | 1.49 |

In Table 3, we compare the performance of the late-fusion MLLMs and their backbone LLMs under the Text-only mode. Qwen-VL-Chat achieves an improvement of 1.49% over Qwen-7B-Chat, while their absolute performance remains below the random baseline. The accuracy of LLMs trained by CogVLM decreases by 4% to 5%, whereas the fluctuation in accuracy for the Llava series is negligible. In summary, the results indicate that current late-fusion approaches do not enhance the LLMs' visual understanding ability, which warrants further exploration.

### 5.3.2 IS THE COMPLEXITY PROPORTIONAL TO THE NUMBER OF CHARACTERS?

We classify test samples into 7 subsets by the number of characters in ASCII art. The results are shown in Fig. 4.

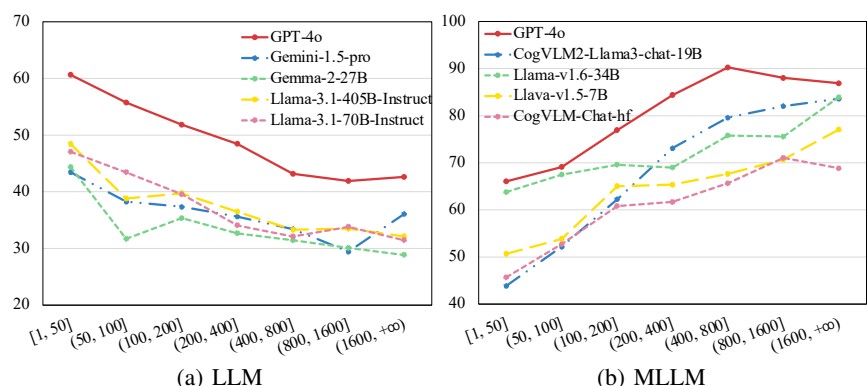

(a) LLM                 (b) MLLM

Figure 4: Micro-accuracy (%) of models on ASCII art with different numbers of characters.

LLMs are proficient in recognizing ASCII art with fewer characters, and they even outperform competitive MLLMs with image inputs, such as CogVLM2-Llama3-chat-19B, on ASCII arts with fewer than 50 characters. In smaller ASCII art, significant features are densely packed within consecutive characters. For instance, the string "() '` ;" captures some major characteristics of a dog in Fig. 1. The results indicate that LLMs excel at capturing the relationship between a concept and some featured combinations of characters. However, as the length of the ASCII art increase, such features are likely to be diluted, and much stronger 2D perception abilities are required.

Conversely, MLLMs are better at recognizing larger ASCII art. Smaller ASCII art tends to be more abstract, where artists try to depict significant features of a concept with few characters. In contrast, larger ASCII art is more similar to real images or posters that MLLMs are trained on. For example, the Spiderman in Fig. 1 shares much more similarity in terms of outline and luminance contrast to a real poster. Nevertheless, MLLM also face challenges on ASCII arts containing more than 1600 characters, as evidenced by the performance drop of both GPT-4o and CogVLM-Chat-hf. This may be due to the fact that larger ASCII art contains more spaces with the same grayscale, providing ineffective or redundant features for MLLMs, thereby aggravating the recognition difficulty.

In summary, LLMs are adept at understanding short and abstract art, and MLLMs are proficient in interpreting longer and more detailed art, which are mainly influenced by the characteristics of the input modality (More in Appendix F). Although different modalities have strengths with various forms of ASCII art, late fusion strategies fail to combine them effectively, as showed in Sec. 5.3.1.

### 5.3.3 How do models perform on different categories?

Models' performances across the 7 different classes are shown in Fig. 5. LLMs trained purely on text corpus perform better at recognizing ASCII arts belonging to the "objects" class. MLLMs given image inputs show consistent improvement in recognizing "travel & places" over LLMs compared to other classes relatively. Moreover, all models struggle with ASCII art referring to "symbols", which comprise different logos and astrology symbols. MLLMs actually perform quite well at recognizing well-known logos, such as Apple and Linux, where GPT-4o achieves 97.96% macro-accuracy and CogVLM2-Llama3-Chat-19B gets 91.16%. However, their performance drops dramatically on relatively niche astrology symbols. Nevertheless, it is simple for both LLMs and MLLMs to answer the question "Can you show me some astrology symbols?". Existing models tend to use rare Unicode characters or emojis to explain the symbols, but cannot understand the visual semantics embedded in those symbols flexibly. More cases can be found in Appendix J.

### 5.4 Can Supervised Fine-tuning elicit models' visual perception capability?

We fine-tune Llama-3.1-8B-Instruct and Llava-v1.6-mistral-7B using ASCIITUNE constructed in Sec. 3.3. The LLM is trained solely with the Text-only data setting, while the MLLM is trained under four different settings: Text-only, Image-only, Text-Image, and Random. "Random" represents that we uniformly select from the above three modality settings for each input sample. All of the models

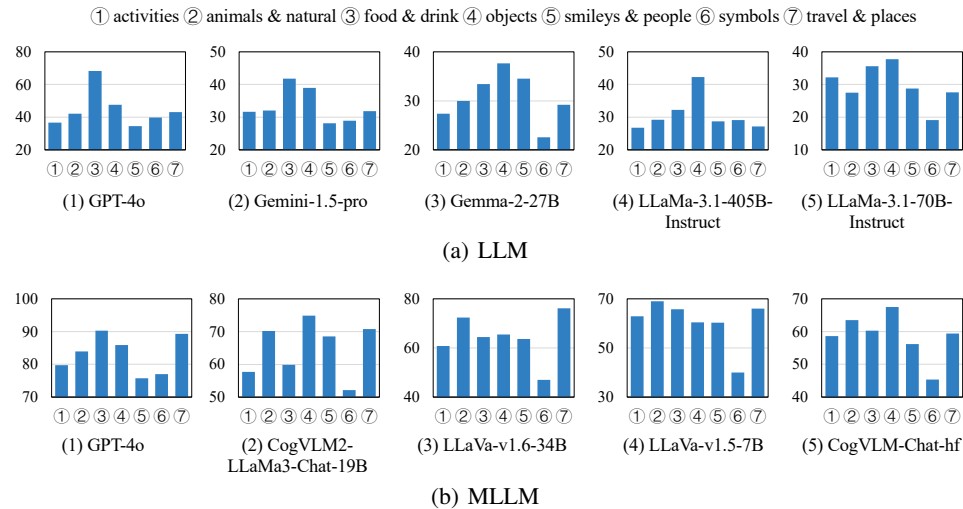

Figure 5: Macro-accuracy (%) of models on recognizing ASCII arts under different classes.

Table 4: Macro-accuracy(%) of the model after supervised fine-tuning on different modes of training data. The corresponding performance on ASCIIEVAL are shown in the last three columns.

| Model | SFT Data | Text-only | Image-only | Text-Image |
|---|---|---|---|---|
| Llama-3.1-8B-Instruct | Text-only | $27.46_{\uparrow 0.24}$ | - | - |
| Llava-v1.6-mistral-7B | Text-only | $26.49_{\uparrow 0.60}$ | - | - |
| | Image-only | - | $75.58_{\uparrow 14.86}$ | - |
| | Text-Image | $25.50_{\downarrow 0.39}$ | $76.78_{\uparrow 16.06}$ | $76.92_{\uparrow 17.90}$ |
| | Random | $27.19_{\uparrow 1.30}$ | $74.52_{\uparrow 13.80}$ | $74.92_{\uparrow 15.90}$ |

are tuned for 2 epochs with a batch size of 16. The results of the fine-tuned models and comparisons to the original results are shown in Table 4.

Models with pure text inputs don't significantly benefit from fine-tuning on the task-specific dataset. They achieve at most a 1.30% improvement under the Text-only setting, while MLLMs with image inputs gains more than a 10% increase accuracy. Even though models are able to recognize ASCII art strings as analyzed above, the experiments also highlight the limitations of current models.

Moreover, supervised fine-tuning on this dataset helps Llava better leverage the representations from both modalities, as shown by the further improvements of Text-Image results over the Image-only results. The decrease in performance of Llava tested under the Text-only mode indicates that the model tends to gather useful information from the image modality when trained by text-image pairs. Llava trained using the Random setting shows better performance on Text-only samples, though with a compromise on samples with image inputs. Exploring training techniques to make the information among modalities more compatible and to improve the accuracy remains a direction of future work.

## 6 CONCLUSION

In this work, we focus on analyzing and eliciting models' visual perception ability in text strings. We introduce the ASCII art recognition problem, which task models to recognize the concepts depicted by the art conveyed through different modalities. We constructed both test and training data, and conducted comprehensive evaluations with dozens of LLMs and MLLMs followed by supervised fine-tuning. Results pinpoint the weaknesses of current models on this task, highlighting a lack of effective fusion techniques for semantic-equivalent information across different carriers.

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

## A  DATA LICENSE

We express our gratitude to the ASCII artists whose fantastic creations underpin our research. In order to assess the visual perception abilities of models, we made slight modifications to the original ASCII art for the test set ASCIIEval. Meanwhile, we retained the original ASCII art and provided the URL to the data source. It is important to note that our data is licensed under CC BY NC 4.0, which permits only non-commercial use and is intended exclusively for research purposes.

## B  FUTURE DIRECTIONS

In the current work, we majorly devoted our efforts on dataset construction for the ASCII art recognition task, benchmarking the performance of LLMs and MLLMs, and figuring out the limitations of current models.

Based on the results and analysis, we summarized future directions as follows:

**Constructing high-quality training data automatically.** We randomly selected 100 samples from ASCIITune for the quality check and the human annotator achieved only 70% accuracy. This indicates that ASCIITune is much noisier than ASCIIEval (98.33%), pointing out the importance of collecting more training data with higher quality. On the one hand, utilizing the ASCII art synthesis tools to convert image datasets into ASCII art can be considered to enlarge the size of the training data, under the awareness of the style differences between the converted ones and the ones created by artists. On the other hand, more strict filtering strategies should be incorporated, such as verifying the validity of ASCII art with strong MLLMs under the Image-only setting.

**Improving the model architecture.** All of the tested LLMs and MLLMs show the inability to recognize information that can be fully represented in text. One potential reason is the lack of exposure to this type of data. It may be also a result of the structural limitation of current models. As for human beings, we perceive text from the aspects of character sequences and their visual shapes at the same time, while these two aspects are conventionally distinguished into two modalities when being processed by neural models. More flexible processing techniques and architecture among modalities should not only benefit the models' visual perception ability in text strings, but also make the model closer to human beings with more efficient information processing abilities.

**Adjusting the training pipeline.** In this work, we simply did supervised fine-tuning with ASCIITune to improve the models' visual perception ability in text strings. This first attempt only shows effectiveness in Image-only setting, pointing out that superficial instruction tuning is not sufficient for current models. Therefore, we hypothesize that post-training should be considered for injecting related knowledge and gaining better representations. Specifically, LLMs are expected to be post-trained on ASCII art corpus containing more diverse tasks, such as ASCII art generation and ASCII art description, mixed with traditional pre-training corpora. As for MLLMs, besides improving the corresponding backbone LLMs, more flexible usage of ASCII art in both modalities should improve MLLMs representation alignments between modalities during the vision-text alignment stage.

We did some further explorations on improving LLMs' and MLLMs' on ASCIIEVAL with unsupervised training objectives based on data from ASCIITUNE. Specifically, we post-trained the Llama-3.1-8B-Instruct on textual ASCII art from ASCIITUNE where the loss is calculated from each token in the textual ASCII art, and post-train the Llava-v1.6-mistral-7B with (rendered ASCII art, textual ASCII art) input-output-pairs from ASCIITUNE where the loss is only calculated from the tokens in the textual ASCII art. Both models are post-trained with the following hyper-parameters: "lr = 2e-5, batch size = 16, number of epochs = 3", and are fine-tuned for ASCII art recognition after further post-training. The results are shown in Table 5.

The post-trained LLM achieved 27.58%, almost the same as the 27.46% accuracy shown in Table 4. Meanwhile, fine-tuning MLLMs with (rendered ASCII art image, textual ASCII art) input-output pairs from ASCIITune is also not helpful for ASCII art recognition. We offer the following hypothesis for the above observations: In the recognition task, the model is required to understand the semantic concept behind the textual ASCII art. However, using textual ASCII art alone for post-training can not help to bridge the gap between visual information and semantics in language. Besides, training MLLMs to convert an ASCII art image into its string format is merely a super-

Table 5: Macro-accuracy(%) of the model after unsupervised post-training and supervised fine-tuning on different modes of training data. The corresponding performance on ASCIIEVAL are shown in the last three columns. The subscript numbers represent the difference compared to the results in Table 4.

| Model | Post-train Data | SFT Data | Text-only | Image-only | Text-Image |
|---|---|---|---|---|---|
| Llama-3.1-8B-Instruct | textual ASCII art | Text-only | $27.58_{\uparrow 0.08}$ | - | - |
| Llava-v1.6-mistral-7B | (rendered ASCII art, textual ASCII art) pairs | Text-only | $26.99_{\uparrow 0.50}$ | - | - |
| | | Image-only | - | $60.90_{\downarrow 14.68}$ | - |
| | | Text-Image | $26.69_{\uparrow 1.19}$ | $61.67_{\downarrow 15.11}$ | $56.83_{\downarrow 20.09}$ |
| | | Random | $26.25_{\downarrow 0.94}$ | $59.44_{\downarrow 15.08}$ | $57.20_{\downarrow 17.72}$ |

ficial transcribing task. However, the task of ASCII art recognition further necessitates models to understand the visual semantics, i.e., concept, depicted in the ASCII art.

In summary, base on the above experiments, we recognize that an ideal training corpus used prior to the instruction-tuning stage should contain samples that embed the ASCII art in documents or conversations. In this way, the model can gradually gain knowledge of this data format and understand the semantic meaning behind it based on its context. Nevertheless, the ASCIITune proposed in our work is designed for supervised fine-tuning, and is not suitable as a pre-training corpus for the following reasonings: First, it only contains 11K samples specifically for the ASCII art recognition task. Second, the semantic context for each ASCII art is limited. ASCIITune may be used as the seed data for developing a more diverse and high-quality dataset using data synthesis techniques in the future.

**Incorporating more complicated scenarios.** Currently, we only considered the basic type of ASCII art made up of 95 printable fixed-width ASCII characters. Nevertheless, there also exist more fascinating ASCII arts, such as color ASCII art, 3D ASCII art, animated ASCII art, etc. These different kinds of ASCII art are also valuable for understanding LLMs designed for video understanding (He et al., 2024) and 3D modeling (Hong et al., 2023).

## C   PROMPT TEMPLATE

We designed three prompt templates for different input modes:

```
                    Prompt Template for Text-only Input

    Please answer the multi-choice question based on the given
    ↪  ASCII art:

    [ASCII ART]
    {ascii_art}

    [Question]
    What is depicted in the above ASCII art? {choices}

    Answer with the option's letter from the given choices
    ↪  directly.
```

```
                    Prompt Template for Image-only Input

    Please answer the multi-choice question based on the given
    ↪  ASCII art image.

    [ASCII ART]
    <image>

    [Question]
    What is depicted in the above ASCII art? {choices}

    Answer with the option's letter from the given choices
    ↪  directly.
```

```
                    Prompt Template for Image-text Input

    Please answer the multi-choice question based on the given
    ↪  ASCII art in both image and text formats.

    [ASCII ART Image]
    <image>

    [ASCII ART Text]
    {ascii_art}

    [Question]
    What is depicted in the above ASCII art? {choices}

    Answer with the option's letter from the given choices
    ↪  directly.
```

All of the models except Qwen-VL are evaluated based on these prompt templates with minor modifications to adapt to their default settings, especially for the position of the image.

Qwen-VL is more sensitive to prompt templates according our experiments. Therefore, we adapted the above templates into Qwen-VL's original format, which is "Context: ... Question: ... Answer:".

## D    DATA ANALYSIS AND STATISTICS

During the data filtering process, we recognized that some of the ASCII art have multiple interpretations, which can be summarized into two types:

○ The ASCII art itself, as a kind of art form, is abstract and ambiguous. For instance, certain depictions of cats might resemble rats. Regarding these cases, we asked human annotators to remove such unrecognizable and ambiguous art.

○ The ASCII art is rich in content, potentially allowing two interpretations from different aspects. For example, the third ASCII art in Fig. 11, can be interpreted as a beach scene, coconut tree, sunset, etc. Most of the ASCII art in ASCIIEval only contains a single object, and we also tried to remove such ambiguities by carefully designing and adjusting the classification criterion. Ultimately, there are only less than 1.67% ambiguous cases in ASCIIEval, leading to the imperfect performance of human annotators.

Finally, the number of samples and the hierarchical relationship between classes and groups of ASCIIEVAL illustrated in Figure 2 are shown in Table 6.

Table 6: The number of samples under each category.

| Classes | Groups |
|---|---|
| animals & natural (1,122) | animal (870), plant (130), nature (122) |
| objects (777) | object (451), electronics (192), clothing (81), furniture (53) |
| smileys & people (644) | role (199), character (195), body (146), occupation (68), people (36) |
| activities (473) | event (207), sport (126), activity (84), instrument (35), monument (21) |
| travel & places (406) | transportation (123), building (123), places (30) |
| food & drink (66) | food (66) |
| symbols (38) | logo (27), astrology (11) |

The token length of samples under the Text-only mode tokenized by three representative tokenizers is in Table 7. The ASCII art data used in our experiments respects the context length limitation of nowadays models.

Table 7: Statistics of token length by different tokenizers.

| | ASCIIEval | | | ASCIITune | | |
|---|---|---|---|---|---|---|
| | Min | Max | Avg | Min | Max | Avg |
| Llama-3 Tokenizer | 71 | 2,192 | 262.72 | 69 | 3,673 | 215.10 |
| Mistral-v0.1 Tokenizer | 85 | 2,890 | 332.91 | 83 | 4,294 | 267.93 |
| Qwen-2 Tokenizer | 80 | 2,833 | 278.17 | 78 | 3,996 | 273.40 |

# E  DETAILS ABOUT EVALUATED MODELS

For open-source instructed models, we experiment with the following LLMs and MLLMs:

**LLMs.**  **Llama** (Touvron et al., 2023) contains three collections of generative models with different sizes, including Llama-2, Llama-3, and Llama-3.1; **Qwen** (Bai et al., 2023a) is another group of models with instructed verions, including Qwen, Qwen1.5 and Qwen2 series; **Mistral** (Jiang et al., 2024a) includes different versions of instruction fine-tuned models, i.e., Mistral-7B-Instruct-v0.1, v0.2 and v0.3. Besides, Mixtral-8x7B-Instruct-v0.1 and Mixtral-8x22B-Instruct-v0.1 which are pre-trained generative Sparse Mixture of Experts are also compared; **Gemma** (Team, 2024b) is a family of lightweight text-to-text models with instruction-tuned variants. We considered Gemma-2-9B-it and Gemma-2-27B-it.

**MLLMs.**  **Llava** (Liu et al., 2023a) augmented a pre-trained LLM with a pre-trained vision encoder. The vision model's representations are projected into the LLM's representation space with a projection layer, and it is frozen during instruction tuning while the projector and the backbone LLM are updated; **CogVLM** (Wang et al., 2023b) aims at retaining the original capabilities of the LLM while adding visual understanding abilities. Representations from the pre-trained vision transformer encoder are passed through an MLP adapter as the input, and a group of trainable visual expert modules in the attention and FFN layers are introduced into the LLM. All of the parameters except the ones from the original LLM are tuned; **Qwen-VL** (Bai et al., 2023b) proposed a position-aware vision-language adapter for compressing image features. The model is trained through three stages,

i.e., pre-training, multi-task pre-training and supervised fine-tuning; **Chameleon** (Team, 2024a) is a family of early-fusion token-based mixed-modal models, different from the above late-fusion ones.

We implemented all open-source models with fewer than 100B parameters locally while collecting predictions from the other models through API requests [5].

## F    ANALYSIS ON SAMPLES UNDER DIFFERENT ASCII ART SIZES

Based on the length characteristics of different ASCII art, we divided the test set into various subsets, as shown in Table 8.

Table 8: The number of samples with ASCII arts divided by different characteristics.

| #Characters | [1, 50] | (50,100] | (100, 200] | (200, 400] | (400, 800] | (800, 1600] | (1600, +∞) |
|---|---|---|---|---|---|---|---|
| #Samples | 221 | 366 | 546 | 710 | 760 | 618 | 305 |
| #Lines | [1,5] | (5, 10] | (10, 15] | (15, 20] | (20, 25] | (25,+∞) | - |
| #Samples | 414 | 854 | 699 | 534 | 399 | 626 | - |

The performances of LLMs and MLLMs on testing samples grouped by the number of lines contained in the ASCII art are shown in Fig. 6. The trends are similar to those grouped by the number of characters in Sec 5.3.2, i.e., LLMs favor smaller ASCII art while MLLMs prefer larger ASCII art.

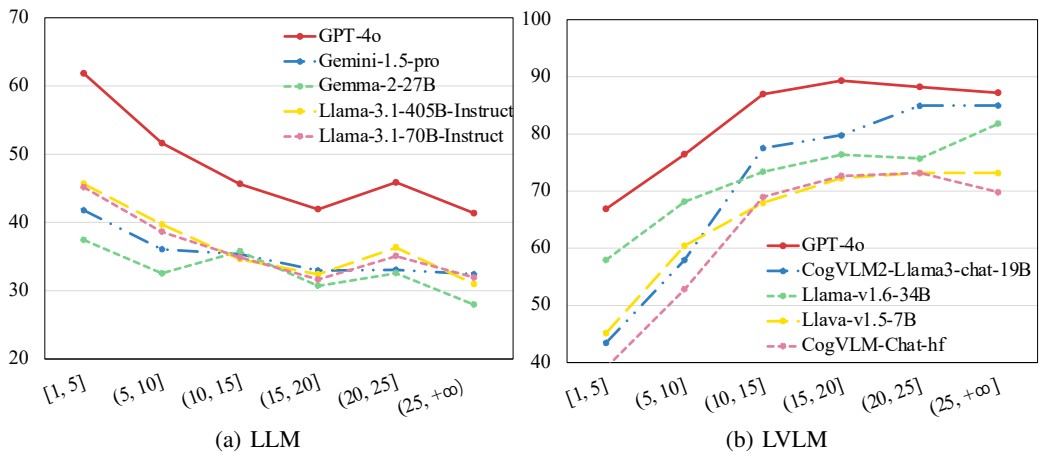

(a) LLM                                    (b) LVLM

Figure 6: Micro accuracy(%) of models on recognizing ASCII arts with different numbers of lines.

We also show the trends of the top 5 MLLMs under Text-only and Text-Image modes respectively in Fig. 7. It reveals that the overall trend in Text-only mode is similar to that of LLMs, indicating that models are easier to be adept at small-sized ASCII art in text format. In contrast, the overall trend in Text-Image mode shares more similarity with the Image-only mode, pointing out the strong bias towards image signals of MLLMs.

## G    PERFORMANCE ON SAMPLES UNDER DIFFERENT CATEGORIES

The models' performance under different groups is shown in Fig. 8. Overall, the performance of MLLMs is more balanced across different categories, except for the drops in "astrology" and "instrument". Meanwhile, LLMs' accuracy fluctuates among different groups, with "electronics", "food" and "object" topping the rank.

---

[5]https://www.together.ai/, https://openai.com/

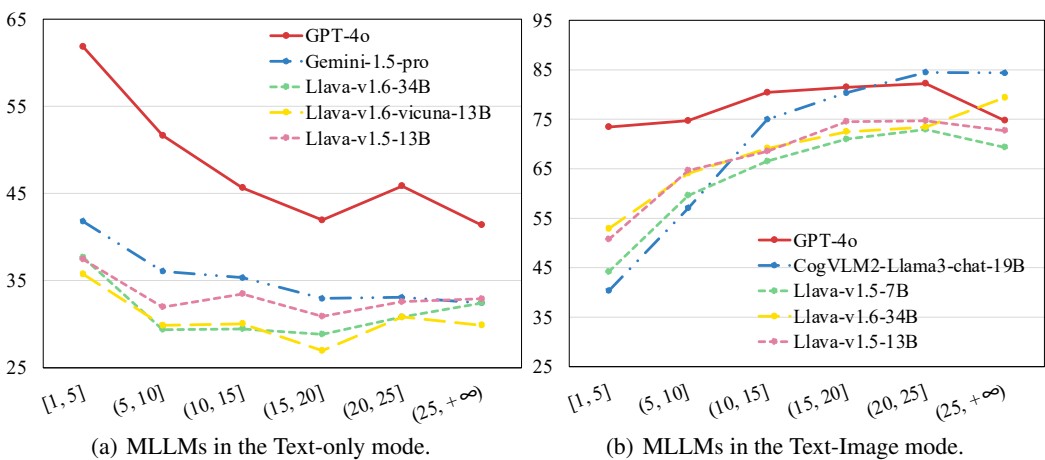

(a) MLLMs in the Text-only mode.   (b) MLLMs in the Text-Image mode.

Figure 7: Micro accuracy(%) of MLLMs on recognizing ASCII arts with different numbers of lines.

## H    SENSITIVITY TO MINOR CHARACTER CHANGES

We randomly removed tokens (other than spaces, "\n" and "\t") from ASCII art and manually checked if the result remained recognizable. Two representative examples are illustrated in Fig. 9. In both cases, the ASCII art remains recognizable when only few characters are removed. However, the first ASCII art becomes progressively indistinguishable as more characters are missing. Meanwhile, the second one just gradually has some additional noise and remains recognizable. This suggests that as the number of characters increases, the importance of each character diminishes as it carries less visual information.

We did more quantitative analysis by sampling 100 cases from ASCIIEval, among which Llava-v1.6-34B provided correct answers under all three test settings. Next, we randomly replaced 1%, 5%, 10%, and 20% of tokens (other than spaces, "\n" and "\t") in the original ASCII art with spaces.

The computed micro-accuracy of Llava-v1.6-34B under different test settings, as well as the human upper bound, are shown in Table 9. Changing the characters in ASCII art will make the recognition task more challenging both for humans and the model, while Human is relatively more robust than Llava-v1.6-34B under different settings.

| Perturbation Ratio | Human | Text-only | Image-only | Text-Image |
|---|---|---|---|---|
| 1% | 99 | 94 | 96 | 96 |
| 5% | 99 | 95 | 91 | 93 |
| 10% | 97 | 91 | 93 | 92 |
| 20% | 94 | 84 | 87 | 83 |

Table 9: The micro-accuracy (%) at different perturbation ratios.

## I    SENSITIVITY WITH DIFFERENT FONTS

In this work, we only considered the traditional ASCII art composed of 95 printable fixed-width ASCII characters. The semantic meaning remains unchanged as long as it is displayed with a fixed-width font. In addition to the "DejaVu Sans Mono" font used in this work, examples of the same ASCII art rendered with 4 different fonts are shown in Fig. 10. All of the dogs are recognizable, with only minor differences. In other words, the multiple-choice questions for ASCII art recognition in ASCIIEVAL remain valid, regardless of the specific fixed-width font used.

Although humans have no difficulty recognizing ASCII art rendered with different fonts, this raises the question of whether MLLMs are sensitive to these variations and show a preference to a specific

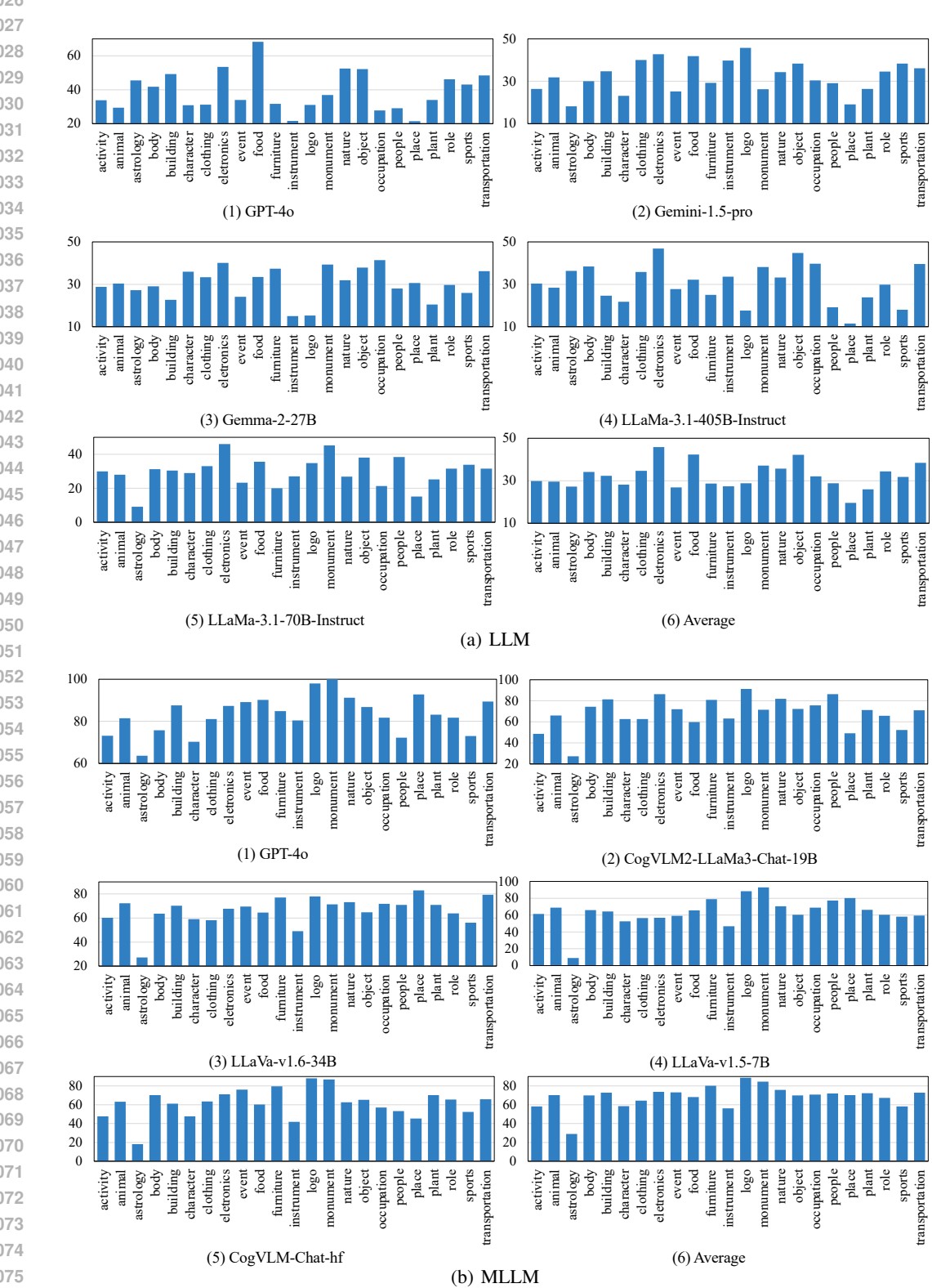

Figure 8: Micro accuracy(%) of models on recognizing ASCII arts in different groups. Average is calculated as the mean of the top 5 models.

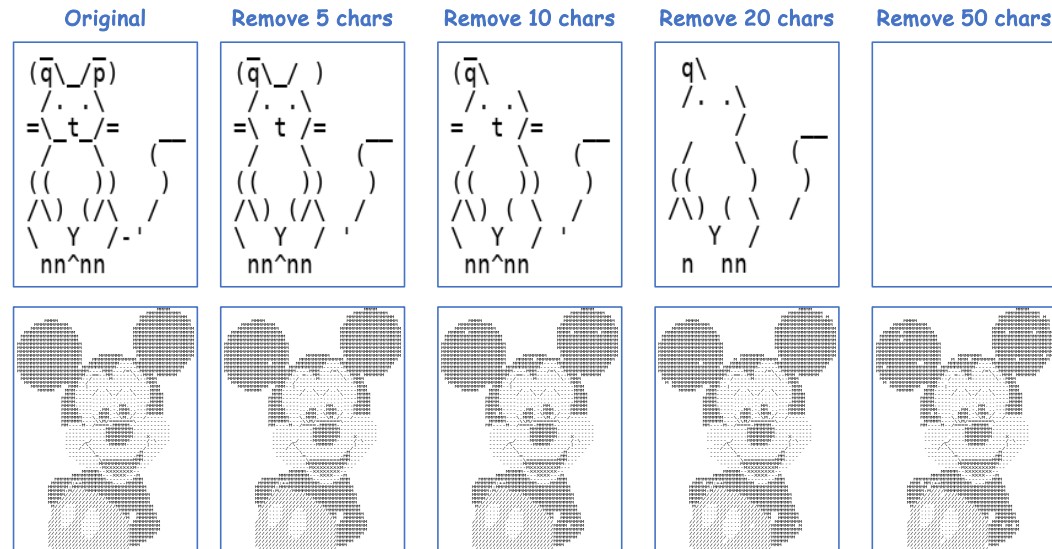

Figure 9: An illustration of removing characters in the ASCII art. "chars" is short for "characters".

Figure 10: An illustration of an ASCII art displayed in different fid-width fonts.

fixed-width font. We take Llava-v1.6-34B as an example and evaluated its performance on ASCII art under both Image-only and Text-Image settings where the images are rendered using 5 different fonts mentioned in Fig. 10. It should be noted that the textual ASCII art is unaffected by font variations, and Llava-v1.6-34B's performance under the Text-only setting is identical to the result in Table 2.

Table 10: Macro-accuracy(%) of Llava-v1.6-34B under Image-only and Text-Image setting with ASCII art rendered by different fix-width fonts.

| Mode | DejaVu Sans Mono | Cascadia Code | Comic Mono | Courier | Fantasque Sans |
|---|---|---|---|---|---|
| Image-only | 65.66 | 63.41 | 66.68 | 63.84 | 66.73 |
| Text-Image | 61.33 | 59.85 | 62.11 | 59.89 | 64.04 |

According to the results in Table 10, MLLMs do face challenges in performing robustly among different text fonts in ASCII art recognition and the performance varies. Nevertheless, its best performance in this table with 66.73% and 64.04% still lags far behind that of GPT-4o with 83.69% and 76.52% under both settings respectively. Moreover, the accuracy under the Text-Image setting is consistently lower than that under the Image-only setting. These observations are same as the results in Sec. 5.2.

On the one hand, how to reduce this sensitivity and improve the MLLMs' robustness is important and worth further exploration. On the other hand, changing the fonts in rendered ASCII art can potentially a useful data augmentation technique for boosting MLLMs' performance on ASCIIEVAL.

## J  CASE STUDIES

We selected seven samples belonging to different classes from ASCIIEVAL and show the cases in Fig. 11 and Fig. 12. The correct answers are shown in red. The top-ranking models are highlighted in yellow if they make correct predictions. Otherwise, they are highlighted in blue with oblique lines.

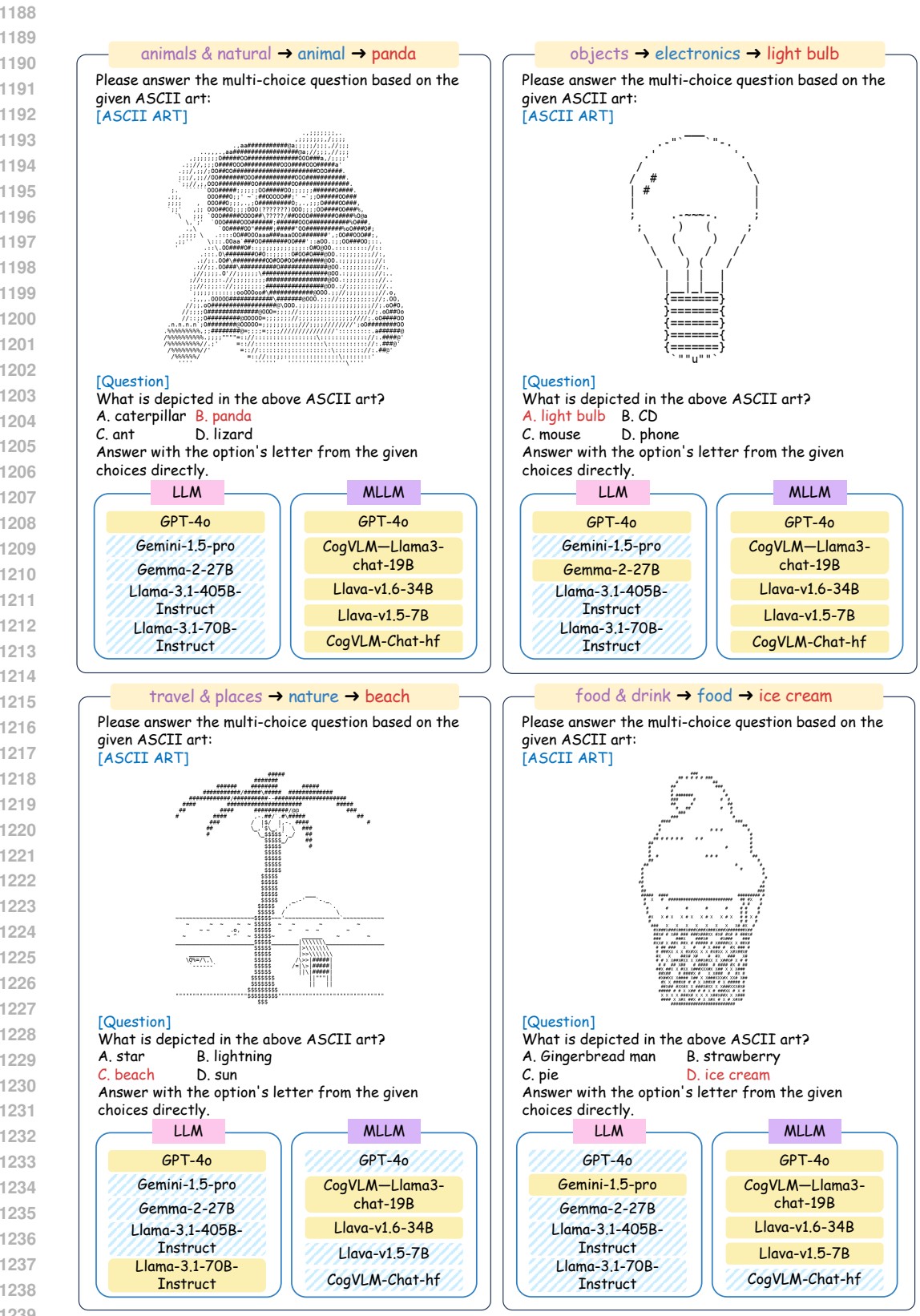

Figure 11: Case studies (Part I).

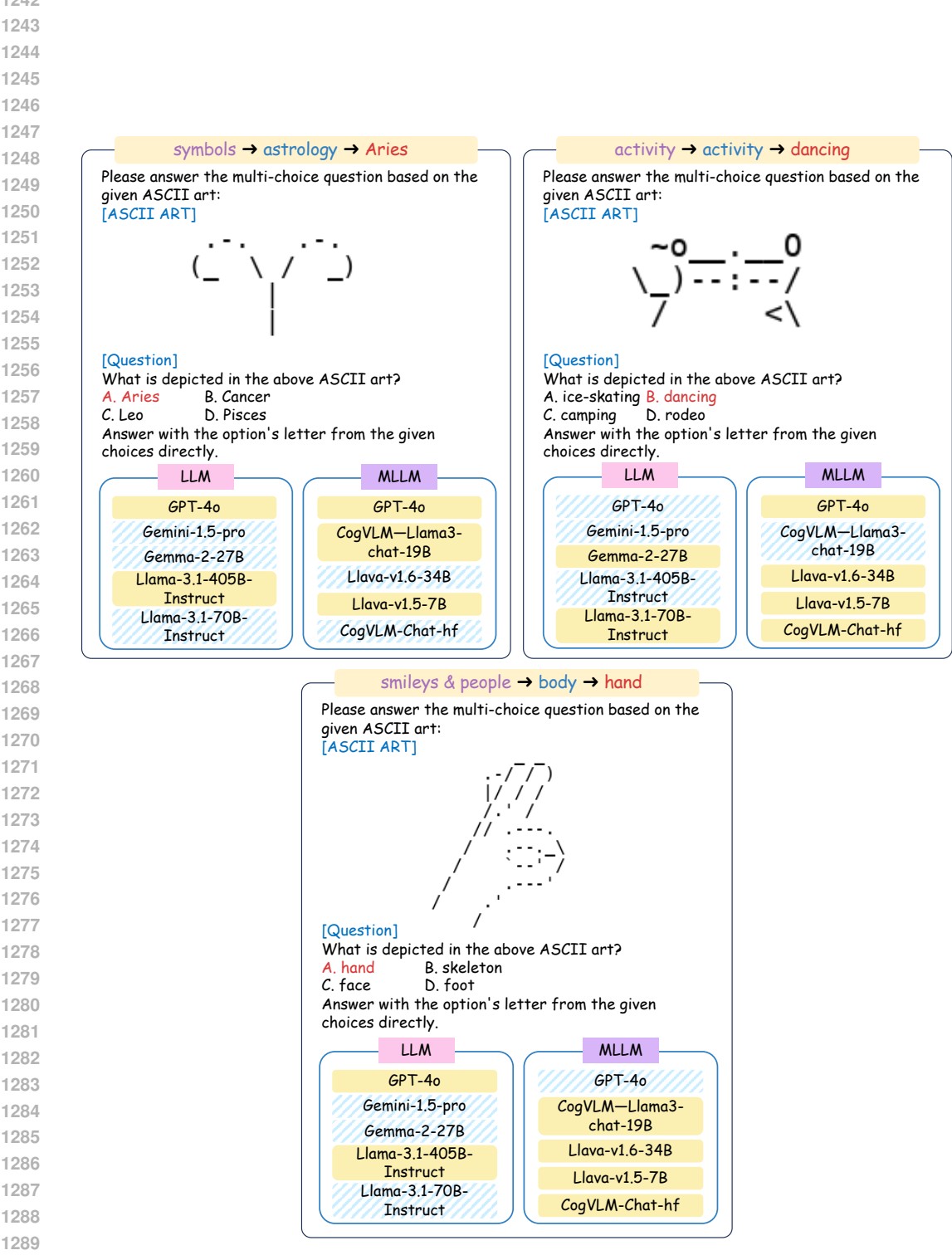

Figure 12: Case studies (Part II).

