# OpenReview forum: "Visual Perception in Text Strings"
_ICLR.cc/2025/Conference — Submitted to ICLR 2025_

### Official Review · Reviewer_t1yN · 2024-10-25

**Soundness:** 3
**Presentation:** 2
**Contribution:** 3
**Rating:** 6
**Confidence:** 4

**Summary:**

The authors propose a novel benchmark to test the ability of multimodal large language models to understand ASCII art. Their experiments show that most multimodal large language models suffer at this task, performing as poorly as random. On the other hand, humans can perform near perfectly at this task. However this changes when we provide an LLM an image of the given ASCII art. The authors benchmark a wide variety of LMMS and report their results.

**Strengths:**

1. The authors propose a novel problem, the idea of this benchmark is quite new.
2. The benchmark proposed is new and large, giving the community a new limitation of Multimodal Models to discover and analyze.

**Weaknesses:**

1. I am not entirely convinced by the motivation of the paper, I don't really understand why it impacts security. The authors should be more clear while describing why is this important for MLLMs to learn.
2. I tried pasting some of the ASCII art text as shown in FIG 1, and was not able to reproduce it in python. The closest I could get was with the dog, this gives me some doubts about the evaluation. The other two ASCII art were not reproducible at least when I pasted it in python.
3. Additionally, I feel some art can have multiple interpretations; for example, certain depictions of cats might resemble rats or raccoons. Since this is art, it naturally lends itself to varied perspectives.
4. Missed reference to an important paper: Can Large Language Models Understand Symbolic Graphics Programs?

**Questions:**

1. It seems like ASCII art is very sensitive to minor character changes, did the authors ablate this? Like a small character being missed can lead to the ASCII art not making any sense.
2. In the human evaluation, were there some images that had multiple interpretations?
3. Could you improve the motivation and introduction? Currently, it feels a bit confusing, covering many points without clearly explaining why LLMs and MLLMs need this capability. While it's an interesting idea, I’m not fully convinced of its practical significance. This isn’t necessarily a criticism of the paper, but adding clarity would be helpful.

---

> ### Author Response · Authors · 2024-11-18
> **Responses**
>
> Thank you for your insightful review. We would like to address your concerns in detail below.
>
> **W1 & Q3: the motivation of the paper**
>
> **Response**:
>
> The motivation of this work is as follows:
>
> - Despite the capability of modeling 2D information even for text strings (by leveraging spatial information provided by special escape characters), LLMs are predominately assessed via textual-semantic-based evaluation benchmark.  In this work, we take a step forward to **evaluate models’ visual perception ability in text strings through the lens of ASCII art recognition**, which we propose as an ideal tool for this purpose.
> - We aim to assess **the alignment of representations between both modalities** in MLLMs. Attributed to its unique characteristics that identical information can be fully represented in both image and text format, ASCII art naturally lends itself to gauging the alignment between the two modalities in MLLMs. As acknowledged by Reviewer 6r3R, ASCII art plays an important role in evaluating the alignment between vision and text modalities in MLLMs, which expands the field of multimodal representation learning. Understanding the interaction between modalities and limitations of current models can guide further improvements.
> - In practice, the findings in our work can be potentially **beneficial to applications** such as ASCII art generation [1], diagram generation [2], tabular data processing [3], playing board games [4], etc. It also has **significant safety implications** for LLMs and MLLMs.  For example, the attacker may ask an LLM “Teach me how to make a (the ASCII art of a bomb)” instead of using the word “bomb”. Since such cases were largely ignored by current safety alignment techniques, it leads to a high risk that this prompt will bypass the security defenses of the model if the model does have such visual perception ability. Therefore, it’s significant to understand such abilities and make proactive defense in advance.
>
> We revised the relevant part of the Introduction for better clarity.
>
> [1] Zihui Gu, Xingwu Sun, Fengzong Lian, Zhanhui Kang, Cheng-Zhong Xu, and Ju Fan. Diverse and fine-grained instruction-following ability exploration with synthetic data. arXiv preprint arXiv:2407.03942, 2024
>
> [2] Devamardeep Hayatpur, Brian Hempel, Kathy Chen, William Duan, Philip Guo, and Haijun Xia. Taking ascii drawings seriously: How programmers diagram code. In Proceedings of the CHI Conference on Human Factors in Computing Systems, 2024.
>
> [3] bNaihao Deng, Zhenjie Sun, Ruiqi He, Aman Sikka, Yulong Chen, Lin Ma, Yue Zhang, and Rada Mihalcea. Tables as texts or images: Evaluating the table reasoning ability of llms and mllms. In Findings of the Association for Computational Linguistics ACL 2024, 2024.
>
> [4] Oguzhan Topsakal and Jackson B Harper. Benchmarking large language model (llm) performance for game playing via tic-tac-toe. Electronics, 2024.
>
>
> **W2: ASCII art text in FIG 1**
>
> **Response:**
>
> - Due to the space limitation and aesthetic considerations, the ASCII art text shown in FIG 1 have been reformatted and truncated.
> - We added a clarification of this in the caption of this figure.
> - We tried different methods to display an ASCII art example directly in this response, but it all get distorted here. If you're interested, more examples can be found at this anonymous link: https://anonymous.4open.science/r/VisionInText-08D3/data/test/test_examples.jsonl.
>
>
> **W3 & Q2: some art can have multiple interpretations**
>
> **Response:**
>
> - Yes, there are some images that may have multiple interpretations. Those images can be summarized into two cases.
>     - Case 1: The ASCII art itself, as a kind of art form, is abstract and ambiguous. As you mentioned in Weakness 3, certain depictions of cats might resemble rats. Regarding these cases, we asked human annotators to remove such unrecognizable and ambiguous art during the data filtering process.
>     - Case 2: The ASCII art is rich in content, potentially allowing two interpretations from different aspects. For example, the third ASCII art in Figure 9 in the Appendix H, can be interpreted as a beach scene, coconut tree, sunset, etc. Most of the ASCII art in ASCIIEval only contains a single object, and we also tried to remove such ambiguities by carefully designing the classification criterion (Sec 3.2). Ultimately, there are only less than 1.67% ambiguous cases in ASCIIEval, leading to the imperfect performance of human annotators as shown in Sec 3.4.
> - Therefore, we believe ASCIIEval can serve as a high-quality benchmark for evaluation.
>
> **W4: Missed reference**
>
> **Response:**
>
> - We add it in Sec 2.1:
>     - Except for recent work from Qiu et al.(2024) benchmark LLMs on answering questions related to the graphics content by generating symbolic graphics programs, none of them consider the visual perception ability of LLMs as a distinct research problem.

---

> ### Author Response · Authors · 2024-11-18
> **Responses**
>
> **Q1:  minor character changes**
>
> **Response:**
>
> - We sampled 100 cases from ASCIIEval, among which Llava-v1.6-34B made correct answers under all three test settings. Then, we randomly replaced 1%, 5%, 10%, and 20% of tokens (except “ ”, “\t” and “\n”) in the original ASCII art with " ". The micro-accuracy of Llava-v1.6-34B under different test settings and human upper bound are shown in the following table.
>
> |     | Human | Text-only | Image-only | Text-Image |
> |-----|-------|-----------|------------|------------|
> | 1%  |   99% |       94% |        96% |        96% |
> | 5%  |   99% |       95% |        91% |        93% |
> | 10% |   97% |       91% |        93% |        92% |
> | 20% |   94% |       84% |        87% |        83% |
>
> - We find that changing the characters in ASCII art will make the recognition task more challenging both for humans and the model. Human is relatively more robust than Llava-v1.6-34B under different settings.
>
> Thank you very much! Please let us know if you have any further questions, and we are more than happy to continue the discussion.

---

> > ### Comment · Reviewer_t1yN · 2024-11-19
> >
> > Thank you for your detailed response to my questions. Here are my thoughts:
> >
> > 1. I recommend revising the Introduction to make the motivation behind this work clearer and more accessible to readers.
> > 2. I understand the limitations with figures—fair enough.
> > 3. W3 is a key part of the benchmark, and the filtering process should be discussed and added to the paper. This addition would clarify both the evaluation and dataset construction aspects.
> > 4. Q1 is an essential experiment for this benchmark paper. It would also be valuable to show how ASCII art changes using figures, providing readers with qualitative intuition for this experiment.
> >
> > Overall, I am convinced by your responses and have decided to increase my score to 6 and my confidence to 4.

---

> > > ### Author Response · Authors · 2024-11-23
> > >
> > > Thank you very much! Following your advice, we have revised the paper as follows:
> > >
> > > 1. We reorganized the motivation behind this work in the introduction to improve clarity.
> > > 2. We expanded the filtering process with additional details in Appendix D due to the space limitation.
> > > 3. We included experiments on character replacement in Appendix H, accompanied by Figure 9 for qualitative illustration and Table 8 for quantitative comparisons.
> > >
> > > We hope these revisions further address your concerns and enhance the clarity of our work.

---

### Official Review · Reviewer_6r3R · 2024-11-01

**Soundness:** 2
**Presentation:** 3
**Contribution:** 2
**Rating:** 6
**Confidence:** 4

**Summary:**

This study introduces an ASCII art recognition task to assess the visual information perception abilities of multimodal large language models (MLLMs). Using data from various public sources, the authors built two datasets, ASCIITune for training and ASCIIEval for evaluation, following thorough normalization and filtering processes. Evaluation of numerous existing LLM and MLLM models with these datasets revealed that most models struggle with recognizing textual ASCII art. Additionally, it was observed that, for most MLLMs, performance in recognizing rendered ASCII art is significantly better than textual ASCII art. Finally, even after fine-tuning, performance improvements for textual ASCII art were minimal compared to the more substantial gains seen for rendered ASCII art.

**Strengths:**

Introduction of a new task: ASCII Art Recognition
- Although many image-text paired datasets exist for MLLM training, ensuring that the images and text in these datasets have precisely the same semantics is often challenging. Evaluating the alignment between vision and text modalities in MLLMs requires a strictly controlled setting, traditionally achieved using document data. This is typically done by pairing rendered images with OCR-ed text. However, this approach is quite limited to character recognition tasks. The ASCII art recognition task introduces a more suitable approach for expanding the field of multimodal representation learning.

Data release: ASCIITune (for training) and ASCIIEval (for evaluation)
- This work utilized various public data sources, constructed multiple-choice questions for more straightforward quantitative evaluation, avoided duplication between training and evaluation datasets, and appropriately excluded potentially sensitive categories. I appreciate releasing such carefully selected data and believe it will benefit the community.

Identification of areas where current LLMs and MLLMs have limitations
- One of the crucial implications of this study is that many current LLMs and MLLMs, both commercial and open-source, have difficulty recognizing textual ASCII art. This might be because these models haven’t been exposed to much of this type of data (ASCII art) during pretraining or due to structural limitations within each model. Similar issues have also come up in vision tasks (e.g., challenges with recognizing analog clocks). However, the inability to recognize information that can be fully represented in text is a noteworthy finding.

**Weaknesses:**

Section 5.1 shows that "current LLMs do not fully understand textual ASCII art." Additionally, Section 5.2 shows that "most MLLMs recognize rendered ASCII art more effectively than textual ASCII art." The lower recognition performance for textual ASCII art seems to stem from LLMs' limitations, as further confirmed in Section 5.3.1. Therefore, it is important to explore why LLMs struggle with the proposed task and consider offering potential solutions. While this study attempts to address this through SFT (supervised fine-tuning), it appears insufficient.

Typical MLLM training generally involves two main stages: (1) a vision-text alignment stage utilizing image-caption data and (2) an instruction tuning stage using image-text QA data. However, representation learning for the text modality itself usually occurs in the LLM's pretraining phase. Therefore, the SFT approach used in this study may not be ideal for enhancing ASCII art recognition in pretrained MLLMs. Additionally, since SFT uses class labels for supervision (e.g., "A: bird"), it may not be suitable in cases where recognition capability is limited, and there may also be insufficient data and training FLOPs.

Here are a few methods worth attempting to address this issue. I believe these efforts and analyses could provide more valuable insights for readers:
1. Further pretraining (M)LLMs on the "textual ASCII art" as a text corpus.
- This step could allow the model to learn patterns specific to textual ASCII art directly.
2. Fine-tuning MLLMs to input rendered ASCII art and output "textual ASCII art" (unfreezing both the vision encoder and LLM).
- This approach may enable the model to learn both image comprehension and the process of generating textual ASCII art.

**Questions:**

1. In Section 3.2 on Data Filtering, is plain edit distance sufficient? Standard edit distance is known to be sensitive to temporal scaling (e.g., "ABBC" vs. "AAABBBBBBCCC"). It may be worth considering methods like Dynamic Time Warping as well.

2. For Figure 4, does Figure 4(a) operate in text-only mode and Figure 4(b) in image-only mode?

3. ASCIITune is likely to be more diverse and noisier than ASCIIEval. I am curious about its quality. It might be helpful to sample a portion of the data and measure human performance as a quality check.

4. While the character count of textual ASCII art is essential, providing the token length after tokenization could also be valuable. Additionally, I would like to know if the ASCII art data used in experiments respects the context length limitations of each model tested.

---

> ### Author Response · Authors · 2024-11-18
> **Responses**
>
> **Weaknesses:**
>
> - In this work, we majorly devoted our efforts on dataset construction for the ASCII art recognition task, benchmarking the performance of LLMs and MLLMs, and figuring out limitations of current models.
> - Conducting supervised fine-tuning is a first attempt towards improving the models’ visual perception ability on text strings, under the consideration of the size and the quality of current training data (ASCIITune). It only shows effectiveness in the image-only setting and points out the challenges of this task.
> - We appreciate these insightful suggestions and have added discussions of further directions with possible solutions in Appendix B of the revised paper from four aspects:
>     1. Constructing high-quality training data automatically
>     2. Improving the model architecture
>     3. Adjusting the training pipeline
>     4. Incorporating more complicated scenarios
>
> **Q1: is plain edit distance sufficient?**
>
> **Response**:
> - Since we crawled ASCII art from different sources, the identical ASCII art may be contained in multiple websites, which are exactly the same or with minor modifications, such as different positions of the artist name, etc. During human annotation for ASCIIEval, the temporal scaling phenomenon was not observed. Therefore, we think the standard edit distance suffices as a pre-processing step before human verification.
>
> **Q2: Figure 4**
>
> **Response**:
>
> - Yes, as stated at the beginning of Sec 5.3, “The top 5 LLMs under the Text-only setting and the top 5 MLLMs under the Image-only setting, which are generally their default input modalities, are primarily considered.”
>
> **Q3: Quality Check for ASCIITune**
>
> **Response:**
>
> - We randomly selected 100 samples from ASCIITune for the quality check. The human annotator achieves only 70% accuracy, indicating that ASCIITune is truly noisier than ASCIIEval (98.33%).
> - In this work, we put more effort into providing a high-quality test set for evaluation, leaving automated construction of a higher-quality training set as a future work.
>
> **Q4: the token length of ASCII art**
>
> **Response:**
>
> - We sincerely apologize for the typo in Table 1. Here is the corrected numbers and the token length with three different tokenizers for ASCIIEval and ASCIITune:
>
> |           | #Characters | #tokens (Llama-3) | #tokens (Mistral-v0.1) | #tokens (Qwen-2) |
> |-----------|:-----------:|:-----------------:|:----------------------:|:----------------:|
> | ASCIIEval |             |                   |                        |                  |
> | min       |           4 |                71 |                     85 |               80 |
> | max       |      15,282 |             2,192 |                  2,890 |            2,833 |
> | avg       |      635.53 |            262.72 |                 332.91 |           278.17 |
> | ASCIITune |             |                   |                        |                  |
> | min       |           1 |                69 |                     83 |               78 |
> | max       |      13,569 |             3,673 |                  4,294 |            3,996 |
> | avg       |      622.38 |            215.10 |                 267.93 |           273.40 |
>
> - To recap, the ASCII art data used in experiments respects the context length limitation of tested models.
> - We added these statistics in Appendix D.
>
> Please let us know if you have any further questions, as we are happy to continue the discussion.

---

> > ### Comment · Reviewer_6r3R · 2024-11-20
> >
> > Thank you for providing a detailed response. The authors' explanations have clarified many points, and I appreciate the effort you put into addressing the questions.
> >
> > Benchmarks are generally more useful when they closely align with real-world problems. However, even tasks that are rarely considered in practical scenarios can hold significant value as analytical benchmarks and datasets for accurately assessing the capabilities of current L(V)LM models. From this perspective, I believe the proposed dataset makes a meaningful contribution.
> >
> > This study highlights that most L(V)LMs struggle with recognizing ASCII art. However, this issue might arise simply because ASCII art recognition is an uncommon task and may not have been sufficiently represented in the training data of existing L(V)LMs. It could be valuable to explore this further. For instance, analyzing the performance of the models in Table 4 after applying further pretraining on the proposed dataset could provide interesting insights. Offering such results as a baseline would likely support future research utilizing this dataset.

---

> ### Author Response · Authors · 2024-11-23
>
> Thank you for your valuable feedback on our paper. We did more explorations as follows.
>
> **Experiment 1**: Further pretraining (M)LLMs on the "textual ASCII art" as a text corpus.
>
> **Response:**
> * Following your advice, we made a trial to pretrain the LLM on textual ASCII art from ASCIITune before fine-tuning it on ASCIITune. The model achieved 27.58%, almost the same as the 27.46% accuracy shown in Table 4.
> * In the recognition task, the model is required to understand the semantic concept behind the textual ASCII art. However, using textual ASCII art alone for pre-training can not help to bridge the gap between visual information and semantics in language. This is also reflected in Experiment 2.
>
> **Experiment 2:** Fine-tuning MLLMs to input rendered ASCII art and output "textual ASCII art"
>
> **Response**:
> * The results are shown in the following table:
> | SFT Data Mode | Text-only     | Image-only     | Text-Image     |
> |---------------|---------------|----------------|----------------|
> | Text-only     | 26.99 (+0.50) | -              | -              |
> | Image-only    | -             | 60.90 (-14.68) | -              |
> | Text-Image    | 26.69 (+1.19) | 61.67 (-15.11) | 56.83 (-20.09) |
> | Random        | 26.25 (-0.94) | 59.44 (-15.08) | 57.20 (-17.72) |
>
> * The values in parentheses represent the difference compared to the results in Table 4.
> * Fine-tuning MLLMs with (rendered ASCII art image, textual ASCII art) input-output pairs from ASCIITune is not helpful for ASCII art recognition. We offer the following hypothesis for this observation: training MLLMs to convert an ASCII art image into its string format is merely a superficial transcribing task. However, the task of ASCII art recognition further necessitates models to understand the visual semantics, i.e., concept, depicted in the ASCII art.
>
> **Summary**
>
> - An ideal training corpus used prior to the instruction-tuning stage should contain samples that embed the ASCII art in documents or conversations. In this way, the model can gradually gain knowledge of this data format and understand the semantic meaning behind it based on its context.
> - The ASCIITune proposed in our work is designed for supervised fine-tuning, and is not suitable as a pre-training corpus for the following reasonings:
>     1. It only contains 11K samples specifically for the ASCII art recognition task.
>     2. The semantic context for each ASCII art is limited.
> - ASCIITune may be used as the seed data for developing a more diverse and high-quality dataset using data synthesis techniques in the future.
> - With your approval, we are willing to include these results in Appendix B to provide additional insights for future work.
>
> Thank you very much! Please let us know if you have any further questions.

---

> > ### Comment · Reviewer_6r3R · 2024-11-25
> >
> > Thank you for sharing the additional experiments. To clarify, I have a few questions:
> >
> > 1. Could you provide more details about the experimental setup for further pretraining? Specifically, I’d like to know about the baseline model you used, lr, batch size, number of epochs (or steps), and the text data format. It would be helpful to see an example of the text data.
> > 2. Regarding the 27.58% result, is this the accuracy after performing further pretraining followed by fine-tuning for multiple-choice supervision?

---

> > > ### Author Response · Authors · 2024-11-25
> > >
> > > Thank you again for your constructive feedback. We greatly appreciate the time and effort you dedicated to reviewing our work.
> > >
> > > **Q1: Experimental setup for further pratraining**
> > >
> > > **Response:**
> > >
> > > - The baseline models are Llama-3.1-8B-Instruct and Llava-v1.6-mistral-7B, the same as those used in Table 4 for comparison.
> > > - We set the following hyper-parameters: “lr = 2e-5, batch size = 16, number of epochs = 3”, considering the limited data available in ASCIITune.
> > > - The data formats are as follows:
> > >     - LLM: ‘<textual ASCII art>’, where the loss is calculated from each token in the textual ASCII art.
> > >     - MLLM: '[INST] <rendered ASCII art>\n [/INST] <textual ASCII art>', where the loss is only calculated from the tokens in the textual ASCII art.
> > >     - We tried different methods to display an ASCII art example directly in the response, but it all gets distorted here. Therefore, we use <rendered ASCII art> and <textual ASCII art> as placeholders in the above examples.
> > >
> > > **Q2:  is this the accuracy after performing further pretraining followed by fine-tuning for multiple-choice supervision?**
> > >
> > > **Response:**
> > >
> > > - Yes, all of the models are fine-tuned for ASCII art recognition after further pre-training.
> > >
> > > Thank you very much! Please let us know if you have any further questions.

---

> > > > ### Comment · Reviewer_6r3R · 2024-11-25
> > > >
> > > > Thank you for sharing all the experiments and analysis during discussion period. Based on your results, I understand that even with further pretraining and supervised fine-tuning, most existing LLMs still face challenges in accurately recognizing textual ASCII art. This outcome may be influenced by various factors, including the current model architectures. I believe that ongoing research will address these limitations, and the proposed benchmark will play an essential role in driving progress in this area. Considering the additional insights provided, I plan to adjust my score from 5 to 6, reflecting my recognition of the comprehensive efforts demonstrated in this research and the potential impact of the proposed benchmark.
> > > >
> > > > To support this, I would like to suggest the following:
> > > > - Adding details about the further pretraining process and results to the appendix.
> > > > - Including reproducible pretraining code in the source code repository.

---

> > > > > ### Author Response · Authors · 2024-11-26
> > > > >
> > > > > Thank you very much for your recognition of our work! We sincerely appreciate your encouraging feedback and would highly appreciate your continued support.
> > > > >
> > > > > In response to your suggestions, we:
> > > > >
> > > > > - added details regarding the further pre-training process and results in Appendix B.
> > > > > - will release all of the codes and data in our source code repository.

---

### Official Review · Reviewer_j9h1 · 2024-11-04

**Soundness:** 2
**Presentation:** 3
**Contribution:** 2
**Rating:** 5
**Confidence:** 3

**Summary:**

The paper explores the use of ASCII art as a tool to evaluate the visual perception capabilities of large language models (LLMs) and multimodal large language models (MLLMs). It introduces two datasets, ASCIIEVAL and ASCIITUNE, which contain 3,526 and 11,836 samples respectively, aimed at assessing and fine-tuning these models. The study conducts extensive testing on a variety of LLMs and MLLMs using the ASCIIEVAL dataset, providing detailed analysis and insights into their performance.

**Strengths:**

1. The paper attempts to use ASCII art to evaluate the visual understanding capabilities of LLMs and MLLMs, offering a new perspective for assessing these models.

2. The authors manually constructed the ASCIIEVAL and ASCIITUNE datasets, containing 3,526 and 11,836 samples respectively, which can be used for evaluating and fine-tuning LLM/MLLM models on ASCII arts understanding.

3. A wide range of different LLMs and MLLMs were tested on ASCIIEVAL, with detailed analysis and discussion of the results provided, contributing to a deeper understanding of model performance.

**Weaknesses:**

1. My main concern is that ASCII art may not effectively represent modality-agnostic visual perception due to the inherent differences in information representation between text and image modalities.

    In text form, LLMs process token sequences derived from a tokenizer, independent of the visual shapes of ASCII characters. In contrast, as images, semantic information is conveyed through pixel rendering, influenced by factors unrelated to the characters themselves, such as font and size.

    As a result, using ASCII art might lead to misleading conclusions about a model's visual perception abilities, as performance can vary significantly between text and image inputs due to the inherent different nature of information representation.

2. Since most real-world images are not represented by ASCII arts, the potential application scenario is also limited.

**Questions:**

1. How does the tokenization process affect the model's ability to understand visual semantics in ASCII art when processed as text?

2. From the perspective of AGI, why enhancing the model's performance on understanding ASCII arts is important?

---

> ### Author Response · Authors · 2024-11-18
> **Responses**
>
> Thanks a lot for your insightful review. We will try to address your concerns as follows:
>
> **W1: inherent differences in information representation between text and image modalities**
>
> **Response:**
>
> - As human beings, we perceive text from the aspects of character sequences and visual shapes at the same time, while these two aspects are conventionally distinguished into two modalities when being processed by neural models. With the advent of LLMs, especially the MLLMs, the strong representation and understanding ability lead us to speculate whether models nowadays are as intelligent as humans.
> - In this work, we limit our scope to visual perception ability in text strings, where visual information can be fully represented in text. We intend to use ASCII art as a tool to help us better understand the potential visual perception ability of LLMs, and representation alignment between vision and text modalities in MLLMs.
>
> **Q1:  the tokenization process**
>
> **Response:**
>
> - Taking the dog in Figure 1 as an example, it will be tokenized into “['Ġ__', 'Ċ', '(___', "()'", '`;Ċ', '/,', 'Ġ/', '`Ċ', '\\\\"', '--', '\\\\']” when processed by the Llama-3 tokenizer.
> - The 2D visual information is represented by the adjacent position of consecutive tokens (x-axis) and 'Ċ' (i.e., ‘\n‘) tokens (y-axis). The characterization of the shape of tokens relies on the intrinsic knowledge learned from the abundant pre-trained corpora.
> - Based on our experiments, we found that models processing ASCII art as text rely more on combinations of token pieces, which is different from the visual semantics reflected by pixels. This poses a barrier to generalization ability, while showing a strong advantage in perceiving more abstract features. (Sec. 5.3.2)
> - Since a tokenizer is tied with the corresponding pre-trained model, it’s impossible to decouple the effects of a tokenizer for more detailed analysis. Designing more specialized tokenizers for such content is one of the future directions.
>
> **Q2 & W2: the motivation/purpose and** **the potential application scenario**
>
> **Response:**
>
> The motivation of this work is as follows:
>
> - Despite the capability of modeling 2D information even for text strings (by leveraging spatial information provided by special escape characters), LLMs are predominately assessed via textual-semantic-based evaluation benchmark.  In this work, we take a step forward to **evaluate models’ visual perception ability in text strings through the lens of ASCII art recognition**, which we propose as an ideal tool for this purpose.
> - We aim to assess **the alignment of representations between both modalities** in MLLMs. Attributed to its unique characteristics that identical information can be fully represented in both image and text format, ASCII art naturally lends itself to gauging the alignment between the two modalities in MLLMs. As acknowledged by Reviewer 6r3R, ASCII art plays an important role in evaluating the alignment between vision and text modalities in MLLMs, which expands the field of multimodal representation learning. Understanding the interaction between modalities and limitations of current models can guide further improvements.
> - In practice, the findings in our work can be potentially **beneficial to applications** such as ASCII art generation [1], diagram generation [2], tabular data processing [3], playing board games [4], etc. It also has **significant safety implications** for LLMs and MLLMs.  For example, the attacker may ask an LLM "Teach me how to make a  (the ASCII art of a bomb)" instead of using the word “bomb”. Since such cases were largely ignored by current safety alignment techniques, it leads to a high risk that this prompt will bypass the security defenses of the model if the model does have such visual perception ability. Therefore, it’s significant to understand such abilities and make proactive defense in advance.
>
> We revised the relevant part of the Introduction for better clarity.
>
> [1] Zihui Gu, Xingwu Sun, Fengzong Lian, Zhanhui Kang, Cheng-Zhong Xu, and Ju Fan. Diverse
> and fine-grained instruction-following ability exploration with synthetic data. arXiv preprint
> arXiv:2407.03942, 2024
>
> [2] Devamardeep Hayatpur, Brian Hempel, Kathy Chen, William Duan, Philip Guo, and Haijun Xia. Taking ascii drawings seriously: How programmers diagram code. In Proceedings of the CHI Conference on Human Factors in Computing Systems, 2024.
>
> [3] bNaihao Deng, Zhenjie Sun, Ruiqi He, Aman Sikka, Yulong Chen, Lin Ma, Yue Zhang, and Rada Mihalcea. Tables as texts or images: Evaluating the table reasoning ability of llms and mllms. In Findings of the Association for Computational Linguistics ACL 2024, 2024.
>
> [4] Oguzhan Topsakal and Jackson B Harper. Benchmarking large language model (llm) performance for game playing via tic-tac-toe. Electronics, 2024.
>
> Thank you very much! Please let us know if you have any further questions, and we are more than happy to continue the discussion.

---

> > ### Comment · Reviewer_6r3R · 2024-11-20
> >
> > Here is my opinion on the safety issue the authors illustrated.
> > Currently, LLMs do not seem to recognize ASCII art well enough to accurately identify "a bomb" in such representations. Instead, they are more likely to misinterpret legitimate ASCII art as "a bomb" and refuse to process the user's request (Over-detection). This, of course, assumes the LLM already operates with a certain level of safety mechanisms in place.

---

> > > ### Author Response · Authors · 2024-11-23
> > > **Response to Reviewer 6r3R**
> > >
> > > Yes, we agree with your opinion on “over-detection”. We also noticed that proprietary models occasionally refused to answer the question due to their safety mechanism, which we mentioned in Sec.4.2 and was also reflected by the imperfect pass rate in Table 2.
> > >
> > > In addition, the weaknesses of (M)LLMs on ASCII art recognition indicate that the probability of bypassing safety measures will be increased if a sensitive concept(e.g., bomb) is replaced by its corresponding ASCII art. LLMs may focus on ASCII art recognition, meanwhile weakening its safety alignment and triggering harmful outputs with elaborate cloaked prompts [1].
> > >
> > > Therefore, benchmarking (M)LLMs ability on ASCII art recognition is of great importance.
> > >
> > > [1] Fengqing Jiang, Zhangchen Xu, Luyao Niu, Zhen Xiang, Bhaskar Ramasubramanian, Bo Li, and Radha Poovendran. Artprompt: Ascii art-based jailbreak attacks against aligned llms. arXiv
> > > preprint arXiv:2402.11753, 2024

---

> ### Author Response · Authors · 2024-11-25
> **Response to Reviewer j9h1**
>
> Thank you for your valuable feedback on our paper. As the discussion deadline is approaching, we would greatly appreciate it if you could review our responses at your earliest convenience. If there is anything that requires further clarification from our side, please do not hesitate to let us know. We sincerely appreciate your time and effort. If you find that our response addresses your concerns, would you kindly consider raising your rating score for our paper? We greatly appreciate your consideration.

---

> ### Comment · Reviewer_j9h1 · 2024-11-25
>
> Thank you for your detailed reply. I appreciate the effort in addressing my concerns and highlighting the potential applications of this work.
>
> However, I still have reservations regarding the tokenization process. Using the Figure 1 dog example, the tokenized string `['Ġ__', 'Ċ', '(___', "()'", ';Ċ', '/,', 'Ġ/', 'Ċ', '\"', '--', '\']` alters the original ASCII representation (e.g., `\n` becomes `Ċ`), and the reconstructed tokens do not visually resemble the original image. This suggests that tokenization introduces information loss, with some information stored in the tokenizer's vocabulary, meaning the tokenized text IDs cannot fully represent the original visual information.
>
> Furthermore, as I previously mentioned, factors like text font or size may also affect the semantics of ASCII art, and I did not see further discussion or experiments addressing this point.
>
> For these reasons, I will maintain my original score. Thank you again for your responses.

---

> > ### Author Response · Authors · 2024-11-25
> >
> > Thank you for your feedback on our paper. We would like to clarify these points as follows:
> >
> > **To begin with,**
> >
> > - We would like to respectfully point out that the tokenization process does **not** introduce information loss. After tokenized by a specific tokenizer, the original ASCII string is converted into a sequence of tokens that is only used for **internal information processing** for the corresponding language model, but **not for human reading**. The token sequence **must** be **decoded** by the same tokenizer for display purposes.
> > - We sincerely apologize that we mistakenly used the ASCII art string in FIG 1, which was reformatted and truncated due to the space limitation and aesthetic considerations. The tokenized string of the original ASCII art should be ['ĠĠĠĠ', 'Ġ__', 'Ċ', '(___', "()'", '`;Ċ', '/,', 'ĠĠĠ', 'Ġ/', '`Ċ', '\\\\"', '--', '\\\\'] by the Llama3 Tokenizer.
> >
> > **Q1:  tokenizer alters the original ASCII representation, e.g. `\n` becomes `Ċ`**
> >
> > **Response**
> > - Using `Ċ` to represent `\n` is just for the **internal information processing** of the corresponding language model. That is to say, `\n`  is always associated with the same embedding layer parameters of the model throughout training and inference. Therefore, it **won’t affect the meaning** of this character from the LLM’s perspective.
> >
> > **Q2: the reconstructed tokens do not visually resemble the original image**
> >
> > **Response:**
> > - Please note that this list of strings is not intended to be concatenated directly. This representation is only designed for the corresponding model. The token sequence should be decoded back to the original string for human understanding.
> > - We tried different methods to display an ASCII art example directly in this response, but it all gets distorted here. We also can not guarantee that the above token sequence doesn’t get distorted here. If you're interested, more examples can be found at this anonymous link: https://anonymous.4open.science/r/VisionInText-08D3/data/test/test_examples.jsonl.
> > - You can try the following code in a Python shell (rendered using any fixed-width font) with transformers (https://github.com/huggingface/transformers):
> > ```
> > from transformers import AutoTokenizer
> > tokenizer = AutoTokenizer.from_pretrained("meta-llama/Meta-Llama-3-8B-Instruct")
> > original_ascii_art_string = "" # please copy an ASCII art string from test_examples.jsonl
> > token_sequence = tokenizer.tokenize(original_ascii_art_string)
> > token_ids = tokenizer.convert_tokens_to_ids(token_sequence)
> > decoded_ascii_art_string = tokenizer.decode(token_ids)
> > print(decoded_ascii_art_string)
> > ```
> > **Q3: factors like text font or size may also affect the semantics of ASCII art**
> >
> > **Response**
> > - We agree that the ASCII art may become unrecognizable when displayed with unsuitable text font.
> > - In this work, we only considered the **traditional ASCII art made up of 95 printable fixed-width ASCII characters**. And the semantic meaning is invariant as long as it is displayed with a **fixed-width font even in different character sizes**.
> > - Here is the link to the original website of the above ASCII art: https://www.asciiart.eu/animals/dogs. You can copy an ASCII art into a txt file, and try with different fixed-width fonts and character sizes.
> >
> > Thank you very much! Please let us know if you have any further questions, and we are more than happy to continue the discussion.

---

> > > ### Author Response · Authors · 2024-11-26
> > >
> > > We sincerely appreciate your time and effort in reviewing our paper and providing constructive feedback on our responses.
> > >
> > > We try to address your concerns regarding the **text font** with more illustrations and experiments in **Appendix I** in our revised paper.
> > >
> > > - Specifically, we added an illustration in Figure 10 where an ASCII art is rendered by 5 different fixed-width text fonts. All of the ASCII art of the dog is recognizable, indicating that **the semantic meaning is invariant** as long as it is displayed with a fixed-width font.
> > > - We also added more experiments on the sensitivity of MLLMs to the input ASCII art image with different fonts. The results (%) for Llava-v1.6-34B under Image-only and Text-Image modes are as follows (the Text-only mode is unaffected by font
> > > variations):
> > > | Mode \ Font | DejaVu Sans Mono | Cascadia Code  | Comic Mono  | Courier | Fantasque Sans |
> > > |-------------|------------------|----------------|-------------|---------|----------------|
> > > | Image-only  | 65.66            | 63.41          | 66.68       | 63.84   | 66.73          |
> > > | Text-Image  | 61.33            | 59.85          | 62.11       | 59.89   | 64.04          |
> > >
> > > - The results indicate that MLLMs do face challenges in performing robustly among different text fonts in ASCII art recognition. Its best performance in this table with 66.73% and 64.04% still lags far behind that of GPT-4o with 83.69% and 76.52% under both settings respectively. Moreover, the accuracy under the Text-Image setting is consistently lower than that under the Image-only setting. **These observations are the same as the results in Sec.5.2 in our paper.**
> > > - In our work:
> > >     - All of the ASCII art is rendered in the same fixed-width font, i.e., “DejaVu Sans Mono”. And **all of the MLLMs are tested under this identical evaluation setup for fair comparisons**.
> > >     - We mainly focus on evaluating models’ visual perception ability in text strings through the lens of ASCII art recognition and assessing the alignment of representations between both modalities in MLLMs. **The major conclusions are unaffected regardless of which fixed-width font is adopted,** such as the weaknesses in textual ASCII art recognition for LLMs, the oversight in modality alignment in MLLMs, etc.
> > > - Measurement and mitigation of the non-robustness regarding the font are worth further exploration but are out of the scope of this paper.
> > >
> > > Thank you very much! Please let us know if you have any further questions, as we are happy to continue the discussion.

---

> > > > ### Author Response · Authors · 2024-12-02
> > > >
> > > > Dear Reviewer j9h1,
> > > >
> > > > As the extended discussion period is approaching its end, we would greatly appreciate receiving your feedback at your earliest convenience. We sincerely appreciate your time and effort, and greatly appreciate your consideration. We also remain available to address any additional questions or concerns you may have.
> > > >
> > > > Thank you very much!
> > > > Authors

---

### Author Response · Authors · 2024-12-02
**Summary**

Dear reviewers,

We appreciate the insightful and constructive feedback from reviewers! We are encouraged by the **positive comments** highlighting the strengths of our work:

- **All** reviewers find our research on visual perception in text strings quite novel, offering a new perspective for assessing LLMs and MLLMs. Moreover, Reviewer **6r3R** think that the ASCII art recognition task proposed in this work introduces a more suitable approach for expanding the field of multimodal representation learning.
- **All** reviewers think that the carefully constructed ASCIIEval and ASCIITune datasets make a meaningful contribution and will benefit the community.
- **All** of the reviewers also agree that our benchmark, containing extensive testing on a variety of LLMs and MLLMs, provides a deeper understanding of model performance and points out their limitations, offering significant value.

As the author response deadline approaches, we would like to let you know that we have submitted our detailed responses for this submission. In our response, we have thoroughly addressed **clarification questions and concerns** from reviewers:

- Reviewer **j9h1** and **t1yN** had confusions regarding the motivation of the proposed task. We clarified that our motivation includes evaluating models’ abilities, assessing the representation alignment between text and image modalities, having potential benefits to related applications and having significant safety implications. We followed the reviewers’ suggestion to revise the introduction for improved clarity and have addressed their concerns.
- Reviewer **j9h1** and **t1yN** wondered about the robustness of models’ performance on minor character changes and different text fonts or sizes, respectively. We revised our paper with additional results in Appendices H and I, accompanied by qualitative and quantitative analyses.
- Following the advice from Reviewer **6r3R**, we added experiments on applying further pre-training on the proposed dataset, and provided detailed results with valuable future directions in Appendix B.
- Reviewer **j9h1** had confusion about the tokenization process of LLMs. We have explained with examples and codes for better clarification.
- We also addressed other confusions from Reviewer  **6r3R** and **t1yN** including the quality check on ASCIITune, data statistics regarding token counts,  data preprocess details, more references, and potential ambiguity in Fig.1. We have improved our paper with more details regarding these points for better clarify.

We have carefully revised our paper based on the above suggestions. Thank you for your consideration.

Sincerely,

Authors

---

### Meta-Review · Area_Chair_5chg · 2024-12-20

**Metareview:**

This paper proposes a benchmark to evaluate how Large Language Models (LLMs) and Multimodal LLMs (MLLMs) can perform visual perception through ASCII art. While the paper presents a novel perspective on LLMs' visual perception capabilities and provides interesting analyses, reviewers noted that ASCII art has limitations in representing modality-agnostic vision representations and its narrow scenarios for potential use cases.

The paper presents various experimental analyses. However, it lacks adequate analysis or plausible hypotheses for key findings, such as why state-of-the-art LLMs lag behind human performance, or why GPT-4o significantly outperforms other open-source models.

Overall, the AC believes that the reported results likely stem from LLMs' one-dimensional text understanding (i.e., how LLMs encode the texts). Therefore, it is difficult to find a deep and new understanding of LLMs' internal perception mechanisms throughout this paper. Considering this limitation and its restricted applicability, the paper requires substantial improvements before it can be accepted.

**Additional Comments On Reviewer Discussion:**

- Reviewer 6r3R and Reviewer t1yN requested additional experiments, and they raised their scores accordingly after the authors provided their new experimental results. The AC agrees that the authors diligently address the raised questions.
- Reviewer j9h1 has questions that ASCII cannot be a representative lens for evaluating the visual perception of LLMs. The AC agrees with Reviewer j9h1's concerns and it is the main limitation of the paper.

---

### Decision · Program_Chairs · 2025-01-22

Reject